# Pure Exploration in Asynchronous Federated Bandits

**Zichen Wang**[1]     **Chuanhao Li**[2]     **Chenyu Song**[3]     **Lianghui Wang**[3]     **Quanquan Gu**[4]     **Huazheng Wang**[3]

[1] University of Illinois Urbana-Champaign, `zichenw6@illinois.edu`
[2] Yale University, `chuanhao.li.cl2637@yale.edu`
[3] Oregon State University, {`songchen,wangl9,huazheng.wang`}`@oregonstate.edu`
[4] University of California, Los Angeles, `qgu@cs.ucla.edu`

## Abstract

We study the federated pure exploration problem of multi-armed bandits and linear bandits, where $M$ agents cooperatively identify the best arm via communicating with the central server. To enhance the robustness against latency and unavailability of agents that are common in practice, we propose the first federated asynchronous multi-armed bandit and linear bandit algorithms for pure exploration with fixed confidence. Our theoretical analysis shows the proposed algorithms achieve near-optimal sample complexities and efficient communication costs in a fully asynchronous environment. Moreover, experimental results based on synthetic and real-world data empirically elucidate the effectiveness and communication cost-efficiency of the proposed algorithms.

## 1 INTRODUCTION

Multi-Armed Bandits (MAB) [Auer et al., 2002, Lattimore and Szepesvári, 2020] is a classic sequential decision-making model that is characterized by the exploration-exploitation tradeoff. Pure exploration [Even-Dar et al., 2006, Soare et al., 2014, Bubeck et al., 2009], also known as best arm identification, is an important variant of the MAB problems where the objective is to identify the arm with the maximum expected reward. While most existing bandit solutions are designed under a centralized setting (i.e., data is readily available at a central server), there is increasing interest in federated bandits in terms of regret minimization [Wang et al., 2019, Li et al., 2022, He et al., 2022] and pure exploration [Hillel et al., 2013, Tao et al., 2019, Du et al., 2021] due to the increasing application scale and public concerns about privacy. Specifically, pure exploration for federated bandits considers $M$ agents identifying the best arm collaboratively with limited communication bandwidth,

while keeping each agent's raw data local. In federated bandits, the major challenge is the conflict between the need for timely data/model aggregation for low sample complexity and the need for communication efficiency with decentralized agents. Balancing model updates and communication is vital to efficiently solve the problem.

Prior works on distributed/federated pure exploration [Hillel et al., 2013, Tao et al., 2019, R'eda et al., 2022, Du et al., 2021] all focused on synchronous communication protocols, where all agents simultaneously participate in each communication round to exchange their latest observations with a central server (federated setting) or other agents (distributed setting). However, the synchronous setting cannot enjoy efficient communication in real-world applications due to 1) some agents may not interact with the environment in certain rounds and 2) the communication in a global synchronous setting needs to wait until the slowest agent responds to the server, which incurs a significant latency especially when the number of the agents is large and the communication is unstable.

To address the aforementioned challenges of model updates and communication, we study the *asynchronous* communication for federated pure exploration problem in this paper. We consider both stochastic multi-armed bandit and linear bandit settings. To reduce communication costs, we propose novel asynchronous event-triggered communication protocols where each agent sends local updates to and receives aggregated updates from the server independently from other agents, i.e., global synchronization is no longer needed. This improves the robustness against possible delays and unavailability of agents. Event-triggered communication only happens when the agent has a significant amount of new observations, which reduces communication costs while maintaining low sample complexity.

With the new communication protocols, we proposed two asynchronous federated pure exploration algorithms, Federated Asynchronous MAB Pure Exploration (`FAMABPE`) and Federated Asynchronous Linear Pure Exploration

*Accepted for the 40th Conference on Uncertainty in Artificial Intelligence* (UAI 2024).

(FALinPE) for MAB and linear bandits, respectively. We theoretically analyzed that these algorithms can return $(\epsilon, \delta)$-best arm with an *efficient communication cost*, *efficient switching cost* and *near-optimal sample complexity*, where the returned arm is $\epsilon$ close to the best arm with probability at least $1 - \delta$, known as fixed confidence setting [Gabillon et al., 2012, Soare et al., 2014, Xu et al., 2017]. Moreover, we empirically validated the theoretical results based on synthetic data and real-world data. Experimental results showed that our event-triggered communication strategy can achieve efficient communication cost, and would only moderately affect the sample complexity compared with the synchronous baselines.

## 2 RELATED WORK

**Pure exploration** The pure exploration problem in single-agent scenarios has been extensively explored in works like Mannor and Tsitsiklis [2004], Even-Dar et al. [2006], Bubeck et al. [2009], Gabillon et al. [2011, 2012], Jamieson et al. [2013], Garivier and Kaufmann [2016], Chen et al. [2016], primarily within the multi-armed bandit framework. Subsequently, Soare et al. [2014], Xu et al. [2017], Tao et al. [2018], Kazerouni and Wein [2019], Fiez et al. [2019], Degenne et al. [2020], Jedra and Proutière [2020] extended these investigations to linear bandits. Advancements by Scarlett et al. [2017], Vakili et al. [2021], Zhu et al. [2021a], Camilleri et al. [2021] further expanded the scope to kernelized bandits. However, these algorithms often suffer from prolonged learning processes and reduced efficacy in the face of limited sample budgets. Hence, our study focuses on the federated resolution of the pure exploration problem.

**Distributed/federated pure exploration** The exploration of pure exploration problem in distributed/federated bandits has become a focal point in recent research. Studies by Hillel et al. [2013], Tao et al. [2019], Karpov et al. [2020], Mitra et al. [2021], R'eda et al. [2022], Chen et al. [2022], Reddy et al. [2022] investigated MAB in a synchronous environment, while Du et al. [2021] explored kernelized bandits synchronously. Primarily designed for synchronous settings, these studies often rely on experimental design to extract exploration sequences. However, such algorithms encounter challenges in asynchronous environments, stemming from their reliance on 1) global synchronous communication rounds, 2) advance knowledge of the active agent for each round, and 3) the server and agents possessing prior knowledge of the time index $t$. Our solution addresses these challenges, presenting the inaugural purely asynchronous algorithms for federated pure exploration with fixed confidence.

**Distributed/federated regret minimization** In tandem with pure exploration, Auer et al. [2002], Abbasi-Yadkori et al. [2011], Filippi et al. [2010], Agrawal and Goyal [2012], Chowdhury and Gopalan [2017] pioneered the study of re-gret minimization in single-agent settings. This problem has recently expanded to distributed/federated bandits, with literature focusing on MAB Szörényi et al. [2013], Korda et al. [2016], Wang et al. [2019], Mahadik et al. [2020], Shi et al. [2021], Zhu et al. [2021b], Yang et al. [2021, 2022, 2023], Patel et al. [2023], linear bandits Wu et al. [2016], Wang et al. [2019], Dubey and Pentland [2020], Huang et al. [2021], Li and Wang [2022b], Amani et al. [2022], Huang et al. [2023], Zhou and Chowdhury [2023], kernelized bandits Li et al. [2022], and neural bandits Dai et al. [2022]. However, these works are confined to synchronous settings. In alignment with our approach, Chen et al. [2023], Li and Wang [2022a], He et al. [2022], Li et al. [2023] targeted regret minimization in an asynchronous environment. Despite this alignment, the primary objectives of regret minimization differ significantly from those of pure exploration, and none of the mentioned works directly addresses our specific problem.

## 3 PRELIMINARIES

In this paper, we let $[t] = \{1, ..., t\}$, $\|\mathbf{x}\|$ denotes the Euclidean norm, $\|\mathbf{x}\|_{\mathbf{V}} = \sqrt{\mathbf{x}^\top \mathbf{V} \mathbf{x}}$ denotes the matrix norm, $\log$ denotes the natural logarithm, $\log_2$ denotes the binary logarithm, $\mathbf{I} \in \mathcal{R}^{d \times d}$ denotes the identity matrix, $\mathbf{0}$ denotes the $d$-dimension zero vector or $d \times d$-dimension zero matrix, $\det(\mathbf{V})$ denotes the determinant of the matrix $\mathbf{V} \in \mathcal{R}^{d \times d}$ and $\mathbf{V}^\top$ denotes the transpose of $\mathbf{V}$. Besides, we utilize $x = \Omega(y)$ to denote that there exists some constant $C > 0$ such that $Cy \leq x$, $x = O(y)$ to denote that there exists some constant $C'$ such that $C'y \geq x$, and $\tilde{O}$ to further hide poly-logarithmic terms.

### 3.1 FEDERATED BANDITS

**MAB** We consider the federated asynchronous MAB (similar to Li and Wang [2022a], He et al. [2022]) as follows. There exists a set $\mathcal{M} = \{m\}_{m=1}^M$ of $M$ agents ($M \geq 2$), a central server and a environment $\mathcal{A} = \{k\}_{k=1}^K$ with $K$ arms ($K \geq 2$). In each round $t$, an arbitrary agent $m_t \in \mathcal{M}$ becomes active, pulls an arm $k_{m_t, t} \in \mathcal{A}$, and receives reward $r_{m_t, t}$. The reward of each arm $k \in \mathcal{A}$ follows a $\sigma$-sub-Gaussian distribution with mean $\mu(k)$. Similar to the other papers that studied the pure exploration [Gabillon et al., 2012, Du et al., 2021], we suppose the best arm $k^* = \arg\max_{k \in \mathcal{A}} \mu(k)$ to be unique.

**Linear bandits** Different from the MAB, in the federated asynchronous linear bandits [Li and Wang, 2022a, He et al., 2022], every arm $k$ is associated with a context $\mathbf{x}_k \in \mathcal{R}^d$. In round $t$, if the active agent $m_t \in \mathcal{M}$ pulls an arm $k_{m_t, t} \in \mathcal{A}$, it would receive reward $r_{m_t, t} = \mathbf{x}_{m_t, t}^\top \boldsymbol{\theta}^* + \eta_{m_t, t}$, where $\boldsymbol{\theta}^* \in \mathcal{R}^d$ is the unknown model parameter and $\eta_{m_t, t} \in \mathcal{R}$ denotes the conditionally $\sigma$-sub-Gaussian noise (more details are provided in Lemma 12 in the appendix). Without

loss of generality, we suppose $\|\mathbf{x}_k\| \le 1, \forall k \in \mathcal{A}, \|\boldsymbol{\theta}^*\| \le 1$ and the best arm $k^* = \arg\max_{k \in \mathcal{A}} \mathbf{x}_k^\top \boldsymbol{\theta}^*$ to be unique.

## 3.2 LEARNING OBJECTIVE

This paper focuses on the fixed confidence $(\epsilon, \delta)$-pure exploration problem. The goal of the bandit algorithm is to find an estimated best arm $\hat{k}^* \in \mathcal{A}$ which satisfies

$$\mathcal{P}(\Delta(k^*, \hat{k}^*) \le \epsilon) \ge 1 - \delta \tag{1}$$

with minimum sample complexity. The reward gap parameter satisfies $0 \le \epsilon < 1$ and the probability parameter satisfies $0 < \delta < 1$. The expected reward gap between arm $i$ and $j$ in the MAB and linear bandits are denoted as $\Delta(i,j) = \mu(i) - \mu(j)$ and $\Delta(i,j) = \mathbf{y}(i,j)^\top \boldsymbol{\theta}^*$, respectively, where $\mathbf{y}(i,j) = \mathbf{x}_i - \mathbf{x}_j$ denotes the difference between contexts. The sample complexity is defined as the agents' total number of interactions with the environment, which is denotes as $\tau$.

## 3.3 COMMUNICATION MODEL AND ASYNCHRONOUS ENVIRONMENT

**Communication model** In this paper, we consider a star-shaped communication network [Wang et al., 2019, Li and Wang, 2022a, He et al., 2022], where every agent can only communicate with the server and can not directly communicate with other agents. We define the communication cost $\mathcal{C}(\tau)$ as the *total number of times* that agents upload data to the server and download data from the server in total $\tau$ rounds [Dubey and Pentland, 2020, Li and Wang, 2022a, He et al., 2022], i.e.,

$$
\begin{aligned}
\mathcal{C}(\tau) = \sum_{t=1}^{\tau} &\mathbb{1}\{m_t \text{ uploads data to the server}\} \\
&+ \mathbb{1}\{m_t \text{ downloads data from the server}\}.
\end{aligned}
\tag{2}
$$

**Asynchronous environment** Similar to He et al. [2022], Li et al. [2023], in the asynchronous environment, there is only one active agent $m_t$ (can be an arbitrary agent in $\mathcal{M}$) that interacts with the environment in each round $t$. Besides, except for the initialization steps, only the active agent is allowed to communicate with the server, i.e., independent from other offline agents.

In our setting, it's important to clarify that the variable $t$ specifically represents the round index, indicating the sequence in which agents engage in the bandit problem. Importantly, it doesn't refer to the actual time of agent involvement. Even when multiple agents are involved, such as in data exchange with the server, there remains a discernible order among these participation events within a very short time frame. This means that even if two events occur very close together in time, a distinct sequence is maintained.

As a result, agent participation happens sequentially, based on the index $t$. Our context has a broader scope compared to previous studies on pure exploration federated bandits [Hillel et al., 2013, Du et al., 2021, Reddy et al., 2022]. This difference arises because those settings require all agents to fully participate in each round, while our setting allows for partial participation, allowing any subset of agents to be involved.

**Motivated example** We here provide a piratical example for asynchronous federated pure exploration. Let's consider a sequential experimental design problem, e.g., for drug discovery or chemical synthesis, where our goal is to identify an arm that is $\epsilon$-near optimal (i.e., chemical with desired properties) with high probability. In this problem, we are not concerned about cumulative regret (i.e., the quality of the chemicals tried during the online learning process); instead, we only care about whether we can find the optimal arm in the end, and the corresponding sample complexity and communication cost due to their expensive nature (see the introduction in Hillel et al. [2013], R'eda et al. [2022], Du et al. [2021] for details). Additionally, each laboratory lacks samples (i.e., funding for resource) to complete the task individually, so we need to involve multiple labs to collaborate on the learning task. These requirements motivate people to study federated pure exploration problems. Besides, previous synchronous federated pure exploration algorithms assume every agent (i.e., lab) should participate in the exploration (i.e., do the experiment) in each round and the server can force all the agents to upload their data in synchronization rounds. This is impractical due to some agents may get offline (e.g., they run out of resources), and all other agents should wait until they get online (e.g., collect enough resources), this will significantly reduce the learning speed (see the introduction in Li and Wang [2022a], He et al. [2022], Li et al. [2023] for details). In this paper, we propose two asynchronous federated pure exploration algorithms that do not rely on synchronous assumptions and are more practical for real-world applications.

## 4 ASYNCHRONOUS ALGORITHMS FOR FEDERATED MAB

In this section, we propose the first asynchronous algorithm for the pure exploration problem of federated MAB. As mentioned in Section 1, a key challenge in conducting pure exploration via asynchronous communication is the absence of dedicated synchronous communication rounds where the server can assign arms to explore each agent based on their latest observations. Moreover, there is no guarantee on when or whether an agent would become active again to execute the exploration and report its observations back. This severely hinders the applicability of all existing distributed/federated pure exploration algorithms, whose exploration strategies are based on experimental design [Hillel

et al., 2013, Du et al., 2021, R'eda et al., 2022]. In order to address this challenge, we adopt a fully adaptive exploration strategy, such that each agent separately and asynchronously decides which arm to pull, based on the statistics received from the server in its latest communication. We name the resulting algorithm Federated Asynchronous MAB Pure Exploration (FAMABPE), and its description is given in Algorithm 1.

**FAMABPE algorithm** As illustrated in lines 2-7, Algorithm 1 begins with an initialization step for $K$ rounds, where the $K$ arms are pulled sequentially. Then the agents and the server update their local statistics accordingly. For round $t \geq K+1$, an agent $m_t$ becomes active and computes its empirical best arm $i_{m_t,t}$ and the most ambiguous arm $j_{m_t,t}$, where

$$i_{m_t,t} = \arg\max_{k \in \mathcal{A}} \hat{\mu}_{m_t,t}(k),$$
$$j_{m_t,t} = \arg\max_{k \in \mathcal{A}/\{i_{m_t,t}\}} \hat{\Delta}_{m_t,t}(k, i_{m_t,t}) \quad (3)$$
$$+ \alpha_{m_t,t}^M(i_{m_t,t}, k),$$

based on which, it selects the most informative arm $k_{m_t,t} = \arg\max_{k \in \{i_{m_t,t}, j_{m_t,t}\}} \alpha_{m_t,t}^M(k)$ to pull in round $t$. We define the arm $k$'s reward estimator of the agent as $\hat{\mu}_{m_t,t}(k)$, the estimated reward gap between arm $i$ and $j$ of the agent as $\hat{\Delta}_{m_t,t}(i,j) = \hat{\mu}_{m_t,t}(i) - \hat{\mu}_{m_t,t}(j)$ and the pair $(i,j)$'s exploration bonus of the agents as $\alpha_{m_t,t}^M(i,j) = \alpha_{m_t,t}^M(i) + \alpha_{m_t,t}^M(j)$ (the definition of $\alpha_{m_t,t}^M(k)$ would be provided in Theorem 1). Intuitively, pulling $k_{m_t,t}$ can most decrease $\alpha_{m_t,t}^M(i_{m_t,t}, j_{m_t,t})$ and thus help reduce sample complexity. After observing reward $r_{m_t,t}$ corresponding to $k_{m_t,t}$, $m_t$ checks the communication event in line 11. If the event is true, agent $m_t$ would upload its local reward sum $S_{m_t,t}^{loc}(k)$ and local observation number $T_{m_t,t}^{loc}(k)$, $\forall k \in \mathcal{A}$ to the server. The server then updates its data and estimation

$$\hat{\mu}_{ser,t}(k) = \frac{\hat{\mu}_{ser,t-1}(k)T_{ser,t-1}(k) + S_{m_t,t}^{loc}(k)}{T_{t-1}^{ser}(k) + T_{m_t,t}^{loc}(k)}, \quad (4)$$
$$T_{ser,t}(k) = T_{ser,t-1}(k) + T_{m_t,t}^{loc}(k), \ \forall k \in \mathcal{A}$$

and

$$i_{ser,t} = \arg\max_{k \in \mathcal{A}} \hat{\mu}_{ser,t}(k),$$
$$j_{ser,t} = \arg\max_{k \in \mathcal{A}/\{i_{ser,t}\}} \hat{\Delta}_{ser,t}(k, i_{ser,t}) + \alpha_{ser,t}^M(i_{ser,t}, k),$$
$$B(t) = \hat{\Delta}_{ser,t}(j_{ser,t}, i_{ser,t}) + \alpha_{ser,t}^M(i_{ser,t}, j_{ser,t}),$$
$$(5)$$

where $\hat{\mu}_{ser,t}$ denotes the arm $k$'s reward estimator of the server, $T_{ser,t}(k)$ denotes the arm $k$'s observation number of the server, $\hat{\Delta}_{ser,t}(i,j) = \hat{\mu}_{ser,t}(i) - \hat{\mu}_{ser,t}(j)$ denotes the estimated reward gap of the server, and $\alpha_{ser,t}^M(i,j) = \alpha_{ser,t}^M(i) + \alpha_{ser,t}^M(j)$ (the setup of $\alpha_{ser,t}^M(k)$ is shown in Theorem 1) denotes the pair $(i,j)$'s exploration bonus of

the server. If the breaking index $B(t) \leq \epsilon$, the server would set its estimated best arm $\hat{k}^* = i_{ser,t}$ and terminate the algorithm (which implies $\tau = t$). Otherwise, agent $m_t$ would download $\hat{\mu}_{ser,t}(k)$ and $T_{ser,t}(k)$, $\forall k \in \mathcal{A}$ from the server and update its local data as shown in lines 18-19. More details are shown in the pseudo-code.

**Low switching cost** Different from the previous distributed/federated pure exploration algorithms [Hillel et al., 2013, Du et al., 2021, R'eda et al., 2022], FAMABPE enjoys a low switching cost (i.e., $1/2\mathcal{C}(\tau)$). The definition of the switching cost is the number of times the agent $m \in \mathcal{M}$ updates $k_{m,t}$ [Abbasi-Yadkori et al., 2011, He et al., 2022, Li et al., 2023]. We suppose $t_1$ and $t_2$ are two neighborhood communication rounds of agent $m$, and $\hat{\mu}_{m,t}(k)$ and $T_{m,t}(k)$, $\forall k \in \mathcal{A}$, would remain unchanged from round $t_1 + 1$ to $t_2$ (line 20∼22 in Algorithm 1). This implies $k_{m,t}$, would also remain unchanged. Hence, the switching cost of FAMABPE equals the total communication number.

**Design of communication event** The event-triggered communication strategy of FAMABPE can control the amount of local data that each agent $m \in \mathcal{M}$ hasn't uploaded, i.e., $\sum_{k=1}^{K} T_{m,t}^{loc}(k)$ and the size of the exploration bonuses simultaneously. Note that in our setting, neither the agents nor the server knows the total number of observations in the system, i.e., time index $t$. Therefore, we utilize $\sum_{k=1}^{K} T_{m,t}(k)$ and $\sum_{k=1}^{K} T_{ser,t}(k)$ to establish the exploration bonuses of agents and server, respectively. This requires $\sum_{k=1}^{K} T_{m,t}(k)$ and $\sum_{k=1}^{K} T_{ser,t}(k)$ to be in a desired proportion to $t$ (which is different from Li and Wang [2022a], He et al. [2022], Li et al. [2023]). Besides, when the server terminates the algorithm, some agents may possess data that has not been uploaded to the server. We wish the amount of these data to be small compared with the sample complexity $\tau$ since they have no contribution to identifying $\hat{k}^*$. Our event-triggered communication protocol can efficiently limit the number of the useless samples.

We can show that our proposed FAMABPE algorithm can attain near-optimal sample complexity $\tau$, with a low communication cost $\mathcal{C}(\tau)$, which is given in the following theorem.

**Theorem 1.** *With $\gamma = 1/(2MK)$ and exploration bonuses*

$$\alpha_{m_t,t}^M(k) =$$
$$\sigma\sqrt{\frac{2}{T_{m_t,t}(k)} \log\left(\frac{4K}{\delta}\left((1+\gamma M)\sum_{k=1}^{K} T_{m_t,t}(k)\right)^2\right)}$$
$$\alpha_{ser,t}^M(k) =$$
$$\sigma\sqrt{\frac{2}{T_{ser,t}(k)} \log\left(\frac{4K}{\delta}\left((1+\gamma M)\sum_{k=1}^{K} T_{ser,t}(k)\right)^2\right)},$$
$$(6)$$

*the estimated best arm $\hat{k}^*$ of FAMABPE can satisfy condition*

**Algorithm 1** Federated Asynchronous MAB Pure Exploration (`FAMABPE`)

---

1: **Inputs:** Arm set $\mathcal{A}$, agent set $\mathcal{M}$, triggered parameter $\gamma$ and $(\delta, \epsilon)$
2: **Initialization:**
3: From round 1 to $K$ sequentially pulls arm from 1 to $K$ and receives reward $r_t, \forall t \in [K]$
4: Server sets $\hat{\mu}_{ser,K}(t) = r_t$ and $T_{ser,K}(t) = 1$          ▷ Server initialization
5: **for** $m = 1 : M$ **do**
6:     Agent $m$ sets $\hat{\mu}_{m,K+1}(k) = r_t$, $T_{m,K+1}(k) = 1$ and $T^{loc}_{m,K}(k) = S^{loc}_{m,K}(k) = 0, \forall k \in \mathcal{A}$    ▷ Agents initialization
7: **end for**
8: **for** $t = K+1 : \infty$ **do**
9:     Agent $m_t$ sets $i_{m_t,t}$ and $j_{m_t,t}$ based on (3), pulls arm $k_{m_t,t}$ and receives reward $r_{m_t,t}$     ▷ Sampling rule
10:    Agent $m_t$ sets $S^{loc}_{m_t,t}(k_{m_t,t}) = S^{loc}_{m_t,t-1}(k_{m_t,t}) + r_{m_t,t}$ and $T^{loc}_{m_t,t}(k_{m_t,t}) = T^{loc}_{m_t,t-1}(k_{m_t,t}) + 1$
11:    **if** $\sum_{k=1}^{K}(T_{m_t,t}(k) + T^{loc}_{m_t,t}(k)) > (1+\gamma)\sum_{k=1}^{K} T_{m_t,t}(k)$ **then**
12:       [**Agent** $m_t$ → **Server**] Send $S^{loc}_{m_t,t}(k)$ and $T^{loc}_{m_t,t}(k), \forall k \in \mathcal{A}$ to the server     ▷ Upload data to server
13:       Server updates $\hat{\mu}_{ser,t}(k), T_{ser,t}(k), \forall k \in \mathcal{A}, i_{ser,t}, j_{ser,t}$ and $B(t)$ based on (4) and (5)
14:       **if** $B(t) \le \epsilon$ **then**
15:          Server returns $i_{ser,t}$ as the estimated best arm $\hat{k}^*$ and break     ▷ Stopping rule and decision rule
16:       **end if**
17:       [**Server** → **Agent** $m_t$] Send $T_{ser,t}(k)$ and $\hat{\mu}_{ser,t}(k), \forall k \in \mathcal{A}$ to agent $m_t$     ▷ Download data from server
18:       Agent $m_t$ sets $T_{m_t,t+1}(k) = T_{ser,t}(k)$ and $\hat{\mu}_{m_t,t+1}(k) = \hat{\mu}_{ser,t}(k), \forall k \in \mathcal{A}$
19:       Agent $m_t$ sets $T^{loc}_{m_t,t}(k) = 0$ and $S^{loc}_{m_t,t}(k) = 0, \forall k \in \mathcal{A}$
20:    **else**
21:       Agent $m_t$ sets $T_{m_t,t+1}(k) = T_{m_t,t}(k)$ and $\hat{\mu}_{m_t,t+1}(k) = \hat{\mu}_{m_t,t}(k), \forall k \in \mathcal{A}$
22:    **end if**
23:    Inactive agent $m \ne m_t$ sets $T_{m,t+1}(k) = T_{m,t}(k)$ and $\hat{\mu}_{m,t+1}(k) = \hat{\mu}_{m,t}(k), \forall k \in \mathcal{A}$
24: **end for**

---

*(1) and with probability at least $1 - \delta$ the sample complexity can be bounded by*

$$\tau \le \frac{M + 1/(2K)}{M - 1/2} H^M_\epsilon 2 \log\left(\frac{4K}{\delta}\left(\left(1 + 1/(2K)\right)\Lambda\right)^2\right),$$

*where*

$$H^M_\epsilon = \sum_{k=1}^{K} \frac{\sigma^2}{\max\left(\frac{\Delta(k^*,k)+\epsilon}{3}, \epsilon\right)^2}$$

$$= O\left(\sum_{k=1}^{K} \frac{1}{(\Delta(k^*,k)+\epsilon)^2}\right)$$

*is the problem complexity in the MAB [Gabillon et al., 2012] and*

$$\Lambda = \left(\frac{M + 1/(2K)}{M - 1/2} H^M_\epsilon 4\right)^2 \frac{4K(1 + 1/(2K))}{\delta^{1/2}}.$$

*The communication cost satisfies $\mathcal{C}(\tau) = \tilde{O}(KM)$.*

**Proof sketch of Theorem 1** Proof of Theorem 1 consists of three main components: a) the communication cost $\mathcal{C}(\tau)$; b) the sample complexity $\tau$; c) the estimated best arm satisfies Eq (1). Specifically, to upper bound the total communications cost $\mathcal{C}(\tau)$, we utilize the property of the event-trigger that controls when the agents would communicate with the server (Lemma 1 in the Appendix). To upper

bound the sample complexity $\tau$, we first need to establish the relation between $\sum_{k=1}^{K} T_{ser,t}(k)$ and $\sum_{k=1}^{K} T^{loc}_{m,t}(k)$ based on the event triggered strategy (Lemma 4 in the Appendix). Then, we establish exploration bonuses by $\sum_{k=1}^{K} T_{ser,t}(k)$ and $\sum_{k=1}^{K} T_{m,t}(k), \forall k \in \mathcal{A}, m \in \mathcal{M}$ and bound $T_{ser,\tau}(k)$, $\forall k \in \mathcal{A}$ accordingly (Lemma 2, 5 and 3 in the Appendix). Finally, utilizing the relations of $T_{ser,t}(k)$ and $T^{all}_t(k) = T_{ser,t}(k) + \sum_{m=1}^{M} T^{loc}_{m,t}(k) = \sum_{s=1}^{t} \mathbb{1}\{k_{m_t,t} = k\}$, we can bound $T^{all}_\tau(k), \forall k \in \mathcal{A}$, and $\tau = \sum_{k=1}^{K} T^{all}_\tau(k)$. The guarantee of finding the best arm, i.e., Eq (1), directly follows the property of the breaking index, i.e., if $B(\tau) \le \epsilon$, then $\Delta(k^*, \hat{k}^*) \le \epsilon$ with probability at least $1 - \delta$.

**Remark 1.** *The sample complexity of `FAMABPE` (i.e., $\tau = O(H^M_\epsilon \log(H^M_\epsilon/\delta))$) can match the lower bound of $(\epsilon, \delta)$ pure exploration problem $\Omega(\sum_{k=1}^{K} \log(1/\delta)/(\Delta(k^*, k) + \epsilon)^2)$ (see details in Lemma 1 of Kaufmann et al. [2014]) up to a constant factor. It implies if we run $(\epsilon, \delta)$ pure exploration algorithms on $M$ agents independently with no communication, the sample complexity is $O(M \sum_{k=1}^{K} \log(1/\delta)/(\Delta(k^*, k) + \epsilon)^2)$ and `FAMABPE` can accelerate the learning process $O(M)$ times. In terms of communication cost, `FAMABPE`'s linear dependence on $M$ matches that attained by previous works studying distributed/federated pure exploration under the less challenging synchronous communication environment [Hillel et al., 2013, Karpov et al., 2020, R'eda et al., 2022, Reddy et al., 2022]. Moreover, the factor $K$ is due to the communication*

*event that ensures $\sum_{k=1}^{K} T_{m,t}(k)$ and $\sum_{k=1}^{K} T_{m,t}^{loc}(k)$ are in a desired proportion to $t$. As mentioned in our previous discussion on its design, this is necessary for the asynchronous communication studied in this paper.*

# 5 ASYNCHRONOUS ALGORITHM FOR FEDERATED LINEAR BANDITS

In this section, we further consider the pure exploration problem of federated linear bandits. We propose an algorithm called Federated Asynchronous Linear Pure Exploration (FALinPE), and its description is given in Algorithm 2.

**FALinPE algorithm** Similar to Algorithm 1, FALinPE starts with an initialization step (line 2-7), where each arm is pulled once. Then in each round $t \geq K + 1$, the active agent $m_t$ sets its estimated model parameter $\hat{\boldsymbol{\theta}}_{m_t,t}$, empirical best arm $i_{m_t,t}$ and most ambiguous arm $j_{m_t,t}$ as

$$
\hat{\boldsymbol{\theta}}_{m_t,t} = \mathbf{V}_{m_t,t}^{-1} \mathbf{b}_{m_t,t}, \quad i_{m_t,t} = \arg\max_{k \in \mathcal{A}} \mathbf{x}_k^\top \hat{\boldsymbol{\theta}}_{m_t,t},
$$
$$
j_{m_t,t} = \arg\max_{k \in \mathcal{A}/\{i_{m_t,t}\}} \hat{\Delta}_{m_t,t}(k, i_{m_t,t}) \tag{7}
$$
$$
+ \alpha_{m_t,t}^L(i_{m_t,t}, k)
$$

and pulls the most informative arm $k_{m_t,t}$ (context denotes as $\mathbf{x}_{m_t,t}$). The exploration bonuses of pair $(i, j)$ in the linear case are defined as $\alpha_{m_t,t}^L(i,j) = \|\mathbf{y}(i,j)\|_{\mathbf{V}_{m_t,t}^{-1}} C_{m_t,t}$ and $\alpha_{ser,t}^L(i,j) = \|\mathbf{y}(i,j)\|_{\mathbf{V}_{ser,t}^{-1}} C_{ser,t}$, where the definitions of the scalers $C_{m_t,t}$ and $C_{ser,t}$ are provided in Theorem 2. Besides, the estimated reward gaps between arm $i$ and $j$ are defined as $\hat{\Delta}_{m_t,t}(i,j) = \mathbf{y}(i,j)^\top \hat{\boldsymbol{\theta}}_{m_t,t}$ and $\hat{\Delta}_{ser,t}(i,j) = \mathbf{y}(i,j)^\top \hat{\boldsymbol{\theta}}_{ser,t}$. Agent $m_t$ would update its covariance matrix $\mathbf{V}_{m_t,t}^{loc}$, $\mathbf{b}_{m_t,t}^{loc}$ and $T_{m_t,t}^{loc}(k_{m_t,t})$ as

$$
\mathbf{V}_{m_t,t}^{loc} = \mathbf{V}_{m_t,t-1}^{loc} + \mathbf{x}_{m_t,t} \mathbf{x}_{m_t,t}^\top,
$$
$$
\mathbf{b}_{m_t,t}^{loc} = \mathbf{b}_{m_t,t-1}^{loc} + r_{m_t,t} \mathbf{x}_{m_t,t}, \tag{8}
$$
$$
T_{m_t,t}^{loc}(k_{m_t,t}) = T_{m_t,t-1}^{loc}(k_{m_t,t}) + 1.
$$

FALinPE utilizes a hybrid event-triggered strategy to control the size of the exploration bonus $\alpha_{ser,t}^L(i,j)$ and $\alpha_{m_t,t}^L(i,j)$, and the observation number $\sum_{k=1}^{K} T_{ser,t}(k)$ and $\sum_{k=1}^{K} T_{ser,t}(k)$. If at least one of the two events is triggered, then agent $m_t$ would upload its collected data $\mathbf{V}_{m_t,t}^{loc}$, $\mathbf{b}_{m_t,t}^{loc}$ and $T_{m_t,t}^{loc}(k)$, $\forall k \in \mathcal{A}$ to the server. The server would update its collected data and estimation

$$
\mathbf{V}_{ser,t} = \mathbf{V}_{ser,t-1} + \mathbf{V}_{m_t,t}^{loc},
$$
$$
\mathbf{b}_{ser,t} = \mathbf{b}_{ser,t-1} + \mathbf{b}_{m_t,t}^{loc},
$$
$$
T_{ser,t}(k) = T_{ser,t-1}(k) + T_{m_t,t}^{loc}(k), \; \forall k \in \mathcal{A}, \tag{9}
$$
$$
\hat{\boldsymbol{\theta}}_{ser,t} = V_{ser,t}^{-1} b_{ser,t}
$$

and set $i_{ser,t}$, $j_{ser,t}$, and the breaking index $B(t)$ as

$$
i_{ser,t} = \arg\max_{k \in \mathcal{A}} \mathbf{x}_k^\top \hat{\boldsymbol{\theta}}_{ser,t},
$$
$$
j_{ser,t} = \arg\max_{k = \mathcal{A}/\{i_{ser,t}\}} \hat{\Delta}_{ser,t}(k, i_{ser,t}) + \alpha_{ser,t}^L(i_{ser,t}, k)
$$
$$
B(t) = \hat{\Delta}_{ser,t}(j_{ser,t}, i_{ser,t}) + \alpha_{ser,t}^L(i_{ser,t}, j_{ser,t}).
$$
$$\tag{10}$$

If the breaking index $B(t) > \epsilon$, the server would return $\mathbf{V}_{ser,t}$, $\mathbf{b}_{ser,t}$ and $T_{ser,t}(k)$, $\forall k \in \mathcal{A}$ to the user. FALinPE would repeat the above steps until $B(\tau) \leq \epsilon$.

**Design of communication events of FALinPE** Similar to FAMABPE, FALinPE also enjoys a low switching cost (i.e, $1/2\mathcal{C}(\tau)$). Besides, the hybrid event-triggered strategy can simultaneously control the size of $\mathbf{V}_{m_t,t}^{loc}$, $\sum_{k=1}^{K} T_{m_t,t}^{loc}(k)$ and the exploration bonuses. Note that Min et al. [2021] also utilize a hybrid event-triggered communication protocol to achieve a similar goal, but for learning stochastic shortest path with linear function approximation. The exploration bonuses in the linear case are not only related to $t$ but also related to covariance matrices. Therefore, different from the communication protocol in the MAB, the event-triggered communication protocol in the linear bandits is additionally required to keep $\mathbf{V}_{m_t,t}$ and $\mathbf{V}_{ser,t}$ in a desired proportion to the global covariance matrix $\lambda \mathbf{I} + \sum_{s=1}^{t} \mathbf{x}_{m_t,t} \mathbf{x}_{m_t,t}^\top$.

**Arm selection strategy** To minimize the sample complexity $\tau$. We hope every agent can pull the most informative arm $k_{m_t,t}$ to reduce the exploration bonus $\alpha_{m_t,t}^L(i_{m_t,t}, j_{m_t,t})$ as fast as possible. The arm selection strategy of Algorithm 2 ensures active agent $m_t$ to pull $k_{m_t,t}$ to most decrease the matrix norm $\|\mathbf{y}(i_{m_t,t}, j_{m_t,t})\|_{\mathbf{V}_{m_t,t}^{-1}}$ (and also $\alpha_{m_t,t}^L(i_{m_t,t}, j_{m_t,t})$). Different from the MAB, in the linear case we can not directly find $k_{m_t,t}$, and need to derive it with a linear programming [Xu et al., 2017], it yields

$$
k_{m_t,t} = \arg\min_{k \in \mathcal{A}} \frac{T_{m_t,t}(k)}{p_k^*(\mathbf{y}(i_{m_t,t}, j_{m_t,t}))}, \tag{11}
$$

where $p^*(\cdot)$ is defined as follows

$$
p_k^*(\mathbf{y}(i_{m_t,t}, j_{m_t,t})) = \frac{w_k^*(\mathbf{y}(i_{m_t,t}, j_{m_t,t}))}{\sum_{s=1}^{K} |w_s^*(y(i_{m_t,t}, j_{m_t,t}))|} \tag{12}
$$

and

$$
\mathbf{w}^*(\mathbf{y}(i_{m_t,t}, j_{m_t,t})) = \arg\min_{\mathbf{w} \in \mathcal{R}^d} \sum_{k=1}^{K} |w_k|
$$
$$\tag{13}$$
$$
s.t. \quad \mathbf{y}(i_{m_t,t}, j_{m_t,t}) = \sum_{k=1}^{K} w_k \mathbf{x}_k.
$$

The notation $w_k^*(\mathbf{y}(i_{m_t,t}, j_{m_t,t}))$ denotes the $k$-th element of vector $\mathbf{w}^*(\mathbf{y}(i_{m_t,t}, j_{m_t,t}))$. Besides, the optimal value of programming (13) denotes as $\rho(\mathbf{y}(i,j)) =$

---

**Algorithm 2** Federated Asynchronous Linear Pure Exploration (`FALinPE`)

---

1: **Inputs:** Arm set $\mathcal{A}$, agent set $\mathcal{M}$, regularization parameter $\lambda > 0$, triggered parameter $\gamma_1, \gamma_2$, and $(\delta, \epsilon)$
2: **Initialization:**
3: From round 1 to $K$ sequentially pulls arm from 1 to $K$ and receives reward $r_t, \forall t \in |K|$
4: Server sets $\mathbf{V}_{ser,K} = \lambda\mathbf{I} + \sum_{t=1}^{K} \mathbf{x}_t\mathbf{x}_t^\top$, $\mathbf{b}_{ser,K} = \sum_{t=1}^{K} \mathbf{x}_t r_t$, $T_{ser,K}(k) = 1, \forall k \in \mathcal{A}$     ▷ Server initialization
5: **for** $m = 1 : M$ **do**
6:      Agent $m$ sets $\mathbf{V}_{m,K+1} = \lambda\mathbf{I} + \sum_{t=1}^{K} \mathbf{x}_t\mathbf{x}_t^\top$, $\mathbf{b}_{m,K+1} = \sum_{t=1}^{K} \mathbf{x}_t r_t$, $T_{m,K+1}(k) = 1, \mathbf{V}_{m,K}^{loc} = \mathbf{0}, \mathbf{b}_{m,K}^{loc} = \mathbf{0}$ and $T_{m,K}^{loc}(k) = 0, \forall k \in \mathcal{A}$     ▷ Agents initialization
7: **end for**
8: **for** $t = K + 1 : \infty$ **do**
9:      Agent $m_t$ sets $\hat{\boldsymbol{\theta}}_{m_t,t}, i_{m_t,t}$ and $j_{m_t,t}$ by (7), pulls $k_{m_t,t}$ by (11) or (14) and receive $r_{m_t,t}$     ▷ Sampling rule
10:      Agent $m_t$ updates $\mathbf{V}_{m_t,t}^{loc}, \mathbf{b}_{m_t,t}^{loc}$ and $T_{m_t,t}^{loc}(k_{m_t,t})$ based on (8)
11:      **if** $\frac{\det(\mathbf{V}_{m_t,t} + \mathbf{V}_{m_t,t}^{loc})}{\det(\mathbf{V}_{m_t,t})} > (1 + \gamma_1)$ **or** $\frac{\sum_{k=1}^{K}(T_{m_t,t}(k) + T_{m_t,t}^{loc}(k))}{\sum_{k=1}^{K} T_{m_t,t}(k)} > (1 + \gamma_2)$ **then**
12:          [**Agent** $m_t \to$ **Server**] Send $\mathbf{V}_{m_t,t}^{loc}, \mathbf{b}_{m_t,t}^{loc}$ and $T_{m_t,t}^{loc}(k), \forall k \in \mathcal{A}$ to the server     ▷ Upload data to server
13:          Server sets $\mathbf{V}_{ser,t}, \mathbf{b}_{ser,t}, T_{ser,t}(k), \forall k \in \mathcal{A}, \hat{\boldsymbol{\theta}}_{ser,t}, i_{ser,t}, j_{ser,t}$ and $B(t)$ based on (9) and (10)
14:          **if** $B(t) \leq \epsilon$ **then**
15:              Server returns $i_{ser,t}$ as the estimated best arm $\hat{k}^*$ and break the loop     ▷ Stopping rule and decision rule
16:          **end if**
17:          [**Server** $\to$ **Agent** $m_t$] Send $\mathbf{V}_{ser,t}, \mathbf{b}_{ser,t}$ and $T_{ser,t}(k), \forall k \in \mathcal{A}$ to agent $m_t$     ▷ Download data
18:          Agent $m_t$ sets $\mathbf{V}_{m_t,t+1} = \mathbf{V}_{ser,t}, \mathbf{b}_{m_t,t+1} = \mathbf{b}_{ser,t}$ and $T_{m_t,t+1}(k) = T_{ser,t}(k), \forall k \in \mathcal{A}$
19:          Agent $m_t$ sets $\mathbf{V}_{m_t,t}^{loc} = \mathbf{0}, \mathbf{b}_{m_t,t}^{loc} = \mathbf{0}$ and $T_{m_t,t}^{loc}(k) = 0, \forall k \in \mathcal{A}$
20:      **else**
21:          Agent $m_t$ sets $\mathbf{V}_{m_t,t+1} = \mathbf{V}_{m_t,t}, \mathbf{b}_{m_t,t+1} = \mathbf{b}_{m_t,t}$ and $T_{m_t,t+1}(k) = T_{m_t,t}(k), \forall k \in \mathcal{A}$
22:      **end if**
23:      Inactive agent $m \neq m_t$ sets $\mathbf{V}_{m,t+1} = \mathbf{V}_{m,t}, \mathbf{b}_{m,t+1} = \mathbf{b}_{m,t}$ and $T_{m,t+1}(k) = T_{m,t}(k), \forall k \in \mathcal{A}$
24: **end for**

---

$\sum_{k=1}^{K} w_k^*(\mathbf{y}(i,j)), \forall i, j \in \mathcal{A}$. However, the programming is computationally inefficient and we also propose to select the arm greedily similar to [Xu et al., 2017]

$$
\begin{aligned}
k_{m_t,t} = \arg\max_{k \in \mathcal{A}} \mathbf{y}(i_{m_t,t}, j_{m_t,t})^\top \\
\times (\mathbf{V}_{m_t,t} + \mathbf{x}_k\mathbf{x}_k^\top)^{-1}\mathbf{y}(i_{m_t,t}, j_{m_t,t}).
\end{aligned} \tag{14}
$$

Although we did not analyze the theoretical property of the greedy arm selection strategy, in the experiment section, we empirically validate that it performs well.

**Theorem 2.** *With* $0 < \lambda \leq \sigma^2\left(\sqrt{1 + \gamma_1 M} + \sqrt{2\gamma_1}M\right)^2 \log(2/\delta)$, $\gamma_1 = 1/M^2$, $\gamma_2 = 1/(2MK)$, *arm selection strategy (11) and 5*

$$
C_{m_t,t} = \sqrt{\lambda} + \left(\sqrt{2\gamma_1}M + \sqrt{1 + \gamma_1 M}\right)
$$
$$
\times \left(\sigma\sqrt{d\log\left(\frac{2}{\delta}\left(1 + \frac{(1 + \gamma_2 M)\sum_{k=1}^{K} T_{m_t,t}(k)}{\min(\gamma_1, 1)\lambda}\right)\right)}\right)
$$
$$
C_{ser,t} = \sqrt{\lambda} + \left(\sqrt{2\gamma_1}M + \sqrt{1 + \gamma_1 M}\right)
$$
$$
\times \left(\sigma\sqrt{d\log\left(\frac{2}{\delta}\left(1 + \frac{(1 + \gamma_2 M)\sum_{k=1}^{K} T_{ser,t}(k)}{\min(\gamma_1, 1)\lambda}\right)\right)}\right),
$$

*the estimated best arm $\hat{k}^*$ of* `FALinPE` *can satisfy condition (1) and with probability at least $1 - \delta$ the sample complexity*

*can be bounded by*

$$
\begin{aligned}
\tau \leq &\frac{M + 1/(2K)}{M - 1/2}\left(\sqrt{2} + \sqrt{1 + \frac{1}{M}}\right)^2 H_\epsilon^L 4\sigma^2 d \\
&\times \log\left(1 + \frac{(1 + 1/(2K))\Lambda^2}{\lambda/M^2}\right) + \Gamma,
\end{aligned}
$$

*where*

$$
H_\epsilon^L = \sum_{k=1}^{K} \max_{i,j \in \mathcal{A}} \frac{\rho(\mathbf{y}(i,j))p_k^*(\mathbf{y}(i,j))}{\max\left(\frac{\Delta(k^*,i)+\epsilon}{3}, \frac{\Delta(k^*,j)+\epsilon}{3}, \epsilon\right)^2}
$$

*is the problem complexity in the linear bandits [Xu et al., 2017],*

$$
\begin{aligned}
\Lambda = &\frac{M + 1/(2K)}{M - 1/2}\left(\sqrt{2} + \sqrt{1 + \frac{1}{M}}\right)^2 H_\epsilon^L 4\sigma^2 d \\
&\times \sqrt{1 + \frac{1 + 1/(2K)}{\lambda/M^2}} + \Gamma
\end{aligned}
$$

*and*

$$
\Gamma = \frac{M + 1/(2K)}{M - 1/2}\left(\sqrt{2} + \sqrt{1 + \frac{1}{M}}\right)^2 H_\epsilon^L 4\sigma^2 d\log\left(\frac{2}{\delta}\right).
$$

*The communication cost satisfies* $\mathcal{C}(\tau) = \tilde{O}\left(\max(M^2 d, MK)\right)$.

**Remark 2.** *As mention in [Xu et al., 2017], the sample complexity of the LinGapE runs by a single agent (i.e., $\tau = \tilde{O}(H_\epsilon^L d)$) can match the lower bound in Soare et al. [2014] up to a constant factor. As shown in the Theorem 2, the sample complexity of FALinPE can also satisfies $\tau = \tilde{O}(H_\epsilon^L d)$ when we select the proper $\lambda$, $\gamma_1$ and $\gamma_2$. Besides, the communication cost of the FALinPE satisfies $\mathcal{C}(\tau) = \tilde{O}(dM^2)$ when $M \geq K/d$, which is the same as the communication cost of Async-LinUCB [Li and Wang, 2022a] and FedLinUCB [He et al., 2022] (both are $\tilde{O}(dM^2)$) in the regret minimization setting. It is worth noting that in our and He et al. [2022]'s setting, the communication between the active agent and server is independent to the offline agent, while in Li and Wang [2022a]'s setting, the algorithm requires a global download section. In addition, the guarantee in Li and Wang [2022a] relies on a stringent regularity assumption on the contexts, while ours and He et al. [2022]'s do not. We claim that the FALinPE can achieve a near-optimal sample complexity and efficient communication cost.*

## 6 EXPERIMENTS

To empirically validate the communication and sample efficiency of FAMABPE and FALinPE, we conduct experiments on both synthetic and real-world dataset. Our algorithms are compared with some baseline algorithms, in which the active agent would share its data with other agents via the server in every round. We compare FAMABPE with single agent UGapEc and synchronous UGapEc [Gabillon et al., 2012] in the MAB setting. Besides, we also compare FALinPE with single agent LinGapE and synchronous LinGapE [Xu et al., 2017]. We clarify that the asynchronous algorithms typically incur larger communication cost than synchronous ones under the same regret/sample complexity guarantee, which is also acknowledged in prior works studying regret minimization [Li and Wang, 2022a, He et al., 2022, Li et al., 2023]. Therefore, the inclusion of synchronous algorithms' mainly serves as a reference showing the performance under the easier synchronous setting. We run the algorithms 10 times and plot their average results.

### 6.1 EXPERIMENTS ON SYNTHETIC DATA

In this section, we report experiments on synthetic dataset for federated MAB and linear bandits.

#### 6.1.1 Experiment Setup

**MAB** We simulate the federated MAB in Section 3.1, with $\sigma = 0.3$, $\delta = 0.05$, $\epsilon = 0$, $K = 5$ and $M = 10$. We sample the optimal arm from the uniform distribution and selectively sample the non-optimal arm to guarantee the

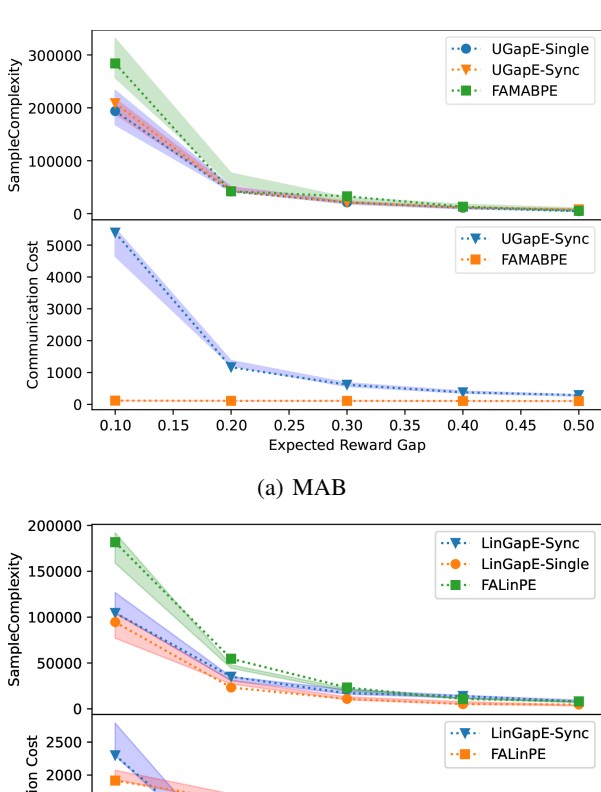

(a) MAB

(b) Linear

Figure 1: Synthetic data: Experimental results for federated MAB and federated linear bandits.

reward gap. For synchronous UGapEc, we set the communication frequency as 100 rounds. At the end of every 100 rounds, the agents would upload their exploration results to the server and download other agents' exploration results from the server. The communication cost of this naive synchronous algorithm is just $C(\tau) = \tau/50$, this is due to there are $\tau/(100M)$ communication episodes and in each episode agents would upload and download data for $2M$ times. The setup of FAMABPE follows Theorem 1 and in each round, the active agent $m_t$ is uniformly sampled from $\mathcal{M}$.

**Linear bandits** Similar to the MAB case, we simulate the federated linear bandits with $d = 5$ and other parameters are the same as the MAB setting. We first sample the model parameter $\theta^*$ from a uniform distribution. Then, we sample the context of the optimal arm and selectively sample non-optimal arms to guarantee the reward gap. The synchronous LinGapE is similar to synchronous UGapEc. The setup of FALinPE follows Theorem 2 and the active agent in the linear case is also uniformly sampled from $\mathcal{M}$.

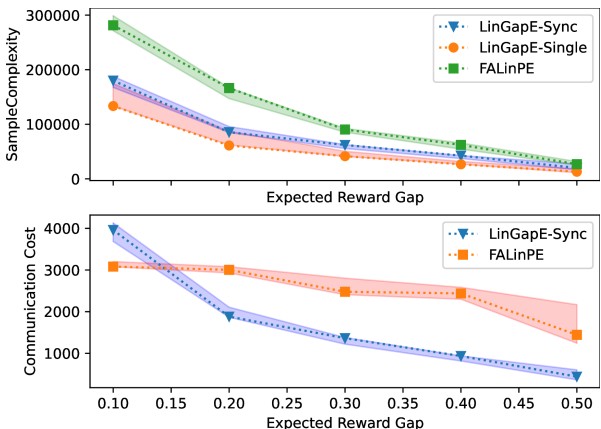

Figure 2: Experimental results on MovieLens for federated linear bandits.

### 6.1.2 Experiment Results

**MAB** The results of federated MAB are shown in Figure 1(a). All algorithms output their estimated best arms $\hat{k}^* = k^*$. We report the sample complexity and communication cost for the reward gap from $0.1$ to $0.5$. We can observe that the single agent which runs UGapEc achieved the smallest sample complexity. In comparison, the synchronous UGapEc would spend a slightly larger sample complexity when the gap equals $0.1$ and spend an almost identical cost when the gap equals $0.2$ to $0.5$. Compared with these baseline algorithms, our FAMABPE had a slightly larger sample complexity and can achieve the lowest communication cost (only took a communication cost of $100$ to $120$ to go from a gap of $0.5$ to a gap of $0.1$). FAMABPE is the only algorithm that can achieve near-optimal sample complexity and efficient communication cost in a fully asynchronous environment.

**Linear bandits** The results of federated linear bandits are provided in Figures 1(b). All algorithms output their estimated best arm $\hat{k}^* = k^*$. Similar to the MAB, we can observe that a single agent which runs LinGapE and synchronous LinGapE achieved the lowest sample complexity. In comparison, FALinPE required a relatively large sample complexity, especially when the gap equals $0.1$. Furthermore, the communication cost of synchronous LinGapE is larger than FALinPE when the gap equals $0.1$. Otherwise, smaller than FALinPE.

### 6.2 EXPERIMENTS ON REAL-WORLD DATA

In this section, we report an additional experiment on real-world dataset for federated linear bandits setting.

### 6.2.1 Experiment Setup

We use the MovieLens 20M dataset [Harper and Konstan, 2016] for the experiment. We follow [Li and Wang, 2022a] to preprocess the data and extract item features. Specifically, we keep users with over $3,000$ observations, which results in a dataset with $54$ users, $26567$ items (movies), and $214729$ interactions. For each item, we extract TF-IDF features from its associated tags and apply PCA to obtain item features with dimension $d = 25$. We consider all items with non-zero ratings as positive feedback (reward $r = 1$), and use ridge regression to learn $\boldsymbol{\theta}^*$ from extracted item features and their $0/1$ rewards. To construct an arm set, we follow the same procedure as the simulation in Section 6 by first sampling an optimal arm and then selectively sampling non-optimal arms to guarantee the reward gap.

The baseline algorithms considered in this section are the same as those in the synthetic data case (Section 6). Besides, we set $d = 25$, $K = 10$, $\epsilon = 0.05$, and other parameters of the federated linear bandits are also identical to Section 6. We report the average results of $10$ runs.

### 6.2.2 Experiment Results

The results of the federated linear bandits are shown in Figures 2. In each run, every algorithm could derive the best arm $k^*$. Similar to the results based on synthetic data, single agent LinGapE and synchronous LinGapE enjoyed the lowest sample complexity, and FALinPE spent a relatively large sample complexity. Besides, according to the tendency, FALinPE's communication cost would be smaller than the synchronous LinGapE's communication cost when the expected reward gap is smaller equals $0.15$. Note that synchronous LinGapE can only work in a synchronous environment, hence, FALinPE is the *only* known federated linear bandit algorithm that can simultaneously achieve near-optimal sample complexity and efficient communication cost in the fully asynchronous environment.

## 7 CONCLUSION

In this paper, we propose the first study on the pure exploration problem of both federated MAB and federated linear bandits in an asynchronous environment. First, we proposed an algorithm named FAMABPE, which can complete the $(\epsilon, \delta)$-pure exploration object of the federated MAB with $\tilde{O}(H_\epsilon^M)$ sample complexity and $\tilde{O}(MK)$ communication cost using a novel event-triggered communication protocol. Then, we improved FAMABPE to FALinPE, which can finish the same object in the linear case with $\tilde{O}(H_\epsilon^L d)$ sample complexity and $\tilde{O}\big(\max(M^2 d, MK)\big)$ communication cost. At the end of the paper, the effectiveness of the offered algorithms was further examined by the numerical simulation based on synthetic data and real-world data. In our future

work, a potential direction is to investigate federated asynchronous pure exploration algorithms with a fixed budget.

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

# A NOTATIONS

| | |
|---|---|
| $\mathcal{A}$ | Arm set |
| $\mathcal{M}$ | Agent set |
| $d$ | Dimension of the model parameter and context |
| $\tau$ | Sample complexity (stopping time) |
| $\mathcal{C}(\tau)$ | Communication cost |
| $k^*$ | Best arm |
| $\hat{k}^*$ | Estimated best arm |
| $m_t$ | Active agent in round $t$ |
| $\eta_{m_t,t}$ | $\sigma$-sub-Gaussian noise |
| $r_{m_t,t}$ | Received reward of agent $m_t$ in round $t$ |
| $k_{m_t,t}$ | Arm pulled by agent $m_t$ in round $t$ |
| $\mu(k)$ | Expected reward of arm $k$ in MAB |
| $\mathbf{x}_k$ | Context of arm $k$ |
| $\mathbf{y}(i,j)$ | Context difference between $\mathbf{x}_i$ and $\mathbf{x}_j$ |
| $\boldsymbol{\theta}^*$ | Model parameter |
| $\delta$ | Probability parameter of the fixed-confidence pure exploration problem |
| $\epsilon$ | Reward gap parameter of the fixed-confidence pure exploration problem |
| $\gamma/\gamma_1/\gamma_2$ | Triggered parameters |
| $\lambda$ | Regularization parameter of the covariance matrix |
| $B(t)$ | Breaking index |
| $\alpha_{m_t,t}^M(k)$ | Exploration bonus of arm $k$ of the agent $m_t$ in FAMABPE |
| $\alpha_{ser,t}^M(k)$ | Exploration bonus of arm $k$ of the server in FAMABPE |
| $\alpha_{m_t,t}^L(i,j)$ | Exploration bonus of pair $(i,j)$ of the agent $m_t$ in FALinPE |
| $\alpha_{ser,t}^L(i,j)$ | Exploration bonus of pair $(i,j)$ of the server in FALinPE |
| $\hat{\boldsymbol{\theta}}_{m_t,t}$ | Estimated model parameter of the agent $m_t$ in FALinPE |
| $\hat{\boldsymbol{\theta}}_{ser,t}$ | Estimated model parameter of the server in FALinPE |
| $i_{m_t,t}$ | Empirical best arm of agent $m_t$ in round $t$ |
| $j_{m_t,t}$ | Most ambiguous arm of agent $m_t$ in round $t$ |
| $\Delta(i,j)$ | Expected reward gap between arms $i$ and $j$ |
| $\hat{\Delta}_{m_t,t}(i,j)$ | Estimated reward gap between arms $i$ and $j$ of the agent $m_t$ |
| $\hat{\Delta}_{ser,t}(i,j)$ | Estimated reward gap between arms $i$ and $j$ of the server |
| $\hat{\mu}_{m_t,t}(k)$ | Estimated reward of arm $k$ of the agent $m_t$ in FAMABPE |
| $\hat{\mu}_{ser,t}(k)$ | Estimated reward of arm $k$ of the server in FAMABPE |
| $T_{m_t,t}(k)$ | Number of observations on arm $k$ that is available to the agent $m_t$ at $\tau$ |
| $T_{ser,t}(k)$ | Number of observations on arm $k$ that is available to the server at $\tau$ |
| $T_{m_t,t}^{loc}(k)$ | Number of observations on arm $k$ has not been uploaded to the server by the agent $m_t$ at $t$ |
| $T_t^{all}(k)$ | Total number of observations on arm $k$ |
| $\mathbf{V}_{m_t,t}$ | Covariance matrix of the agent $m_t$ |
| $\mathbf{V}_{ser,t}$ | Covariance matrix of the server |
| $\mathbf{V}_{m_t,t}^{loc}$ | Covariance matrix has not been uploaded to the server by the agent $m_t$ |
| $\mathbf{V}_t^{all}$ | Global covariance matrix |
| $H_\epsilon^M$ | Problem complexity of the MAB |
| $H_\epsilon^L$ | Problem complexity of the linear bandits |

# B PROOF OF THEOREM 1

For clarity, we here reintroduce the notations used in the proof. Recall that $T_t^{all}(k)$ denotes the total number of arm $k$ be pulled till round $t$, $T_{ser,t}(k)$ denotes the number of observations on arm $k$ that is available to the server at $t$, $T_{m,t}(k)$ denotes the the number of observations on arm $k$ that is available to the agent $m$ at $t$ and $T_{m,t}^{loc}(k)$ denotes the observations on arm $k$ of agent $m$ has not been uploaded to the server at $t$. Besides, $\hat{\Delta}_{ser,t}(i,j)$ and $\hat{\Delta}_{m,t}(i,j)$ denote the estimated reward gap between arm $i$ and $j$ of the agent $m$ and server in round $t$, respectively. Furthermore, we let $\alpha_{m,t}^M(k)$ and $\alpha_{ser,t}^M(k)$ denoting the exploration bonuses of the agent $m$ and server, respectively. Moreover, we define the reward estimator of arm $k$ of the agent $m$ and server as $\hat{\mu}_{m,t}(k)$ and $\hat{\mu}_{ser,t}(k)$, respectively.

**Remark 3** (Global and local data in the federated MAB). *By the design of Algorithm 1, the total number of times arm $k$ has been pulled till round $t$ satisfies $T_t^{all}(k) = T_{ser,t}(k) + \sum_{m=1}^M T_{m,t}^{loc}(k)$, where $T_{ser,t}(k)$ denotes the number of observations on arm $k$ that has been uploaded to the server and $\sum_{m=1}^M T_{m,t}^{loc}(k)$ denotes the total number of observations on arm $k$ that agents $m = 1, 2, \ldots, M$ have not uploaded to the server. Besides, as $T_{m,t}(k), \forall m \in \mathcal{M}, k \in \mathcal{A}$ is downloaded from the server in some round earlier than $t$, we have $T_{m,t}(k) \leq T_{ser,t}(k), \forall k \in \mathcal{A}$ and $\sum_{k=1}^K T_{m,t}(k) \leq \sum_{k=1}^K T_{ser,t}(k)$.*

Detailed proof for the first two components are given in the following two subsections.

## B.1 UPPER BOUND COMMUNICATION COST $\mathcal{C}(\tau)$

**Lemma 1** (Communication cost). *Following the setting of Theorem 1, the total communication cost of FAMABPE can be bounded by*

$$\mathcal{C}(\tau) \leq 2\Big((M + 1/\gamma)\log_2 \tau\Big).$$

*Proof of Lemma 1.* The proof of this Lemma can be divided into two sections, in the first section, we would divide the sample complexity $\tau$ into $\log_2 \tau$ episodes, then we would analyze the upper bound of the communication number of all agents in each episode. We define

$$\mathcal{T}_i = \min\left\{t \in [K+1, \tau], \sum_{k=1}^K T_{ser,t}(k) \geq 2^i\right\}$$

and the set of all rounds into episodes $i$ as $\{\mathcal{T}_i, \mathcal{T}_i+1, ..., \mathcal{T}_{i+1}-1\}$. According to the definition, we have $\sum_{k=1}^K T_{ser,\tau}(k) \leq \tau$, and thus

$$\max\{i \geq 0\} = \log_2\left(\sum_{k=1}^K T_{ser,\tau}(k)\right) \leq \log_2 \tau.$$

We then prove $\forall i \geq 0$, from round $\mathcal{T}_i$ to $\mathcal{T}_{i+1} - 1$, the communication number of all agents can be bounded by $M + 1/\gamma$. We first define the communication number of agent $m$ from $\mathcal{T}_i$ to $\mathcal{T}_{i+1} - 1$ as $\mathcal{N}_m$, the sequence of communication round of agent $m$ from round $\mathcal{T}_i$ to $\mathcal{T}_{i+1} - 1$ as $t_1^m, ..., t_{\mathcal{N}_m}^m$, the communication number of all agents as $L$ and the sequence of all communication rounds from round $\mathcal{T}_i$ to $\mathcal{T}_{i+1} - 1$ as $t_{i,1}, ..., t_{i,L}$. According to the communication condition (line 11 of Algorithm 1), we have $\forall m \in \mathcal{M}, j \in |\mathcal{N}_m|$

$$\sum_{k=1}^K \left(T_{m,t_j^m}(k) + T_{m,t_j^m}^{loc}(k)\right) > (1+\gamma)\sum_{k=1}^K T_{m,t_j^m}(k)$$

$$\sum_{k=1}^K T_{m,t_j^m}^{loc}(k) > \gamma\sum_{k=1}^K T_{m,t_j^m}(k). \tag{15}$$

Then, $\forall m \in \mathcal{M}, j \in |\mathcal{N}_m|/\{1\}$, we have

$$\sum_{k=1}^K T_{m,t_j^m}^{loc}(k) > \gamma\sum_{k=1}^K T_{m,t_j^m}(k) \geq \gamma\sum_{k=1}^K T_{ser,\mathcal{T}_i}(k). \tag{16}$$

The inequality holds due to $T_{m,t_j^m}(k) = T_{t_{j-1}^m}^{ser}(k)$ and $t_{j-1}^m \geq \mathcal{T}_i$ when $j \in |\mathcal{N}_m|/\{1\}$. The above inequality implies $\forall t_{i,l} \geq t_2^{m_{t_{i,l}}}$

$$\sum_{k=1}^{K} \left( T_{ser,t_{i,l}}(k) - T_{ser,t_{i,l-1}}(k) \right) \geq \sum_{k=1}^{K} \left( T_{ser,t_{i,l-1}}(k) + T_{m_{t_{i,l}},t_{i,l}}^{loc}(k) \right) - \sum_{k=1}^{K} T_{ser,t_{i,l-1}}(k)$$

$$= \sum_{k=1}^{K} T_{m_{t_{i,l}},t_{i,l}}^{loc}(k)$$

$$> \gamma \sum_{k=1}^{K} T_{m_{t_{i,l}},t_{i,l}}(k)$$

$$\geq \gamma \sum_{k=1}^{K} T_{ser,\mathcal{T}_i}(k).$$

Finally we can bound $L = \sum_{m=1}^{M} \mathcal{N}_m$

$$\sum_{k=1}^{K} \left( T_{ser,\mathcal{T}_{i+1}-1}(k) - T_{ser,\mathcal{T}_i}(k) \right) = \sum_{l=1}^{L-1} \sum_{k=1}^{K} \left( T_{ser,t_{i,l+1}}(k) - T_{ser,t_{i,l}}(k) \right) \tag{17}$$

$$\geq \gamma \sum_{m=1}^{M} (\mathcal{N}_m - 1) \sum_{k=1}^{K} T_{ser,\mathcal{T}_i}(k).$$

The last inequality holds owing to (15) and (16). With the definition of the episode, we have $\sum_{k=1}^{K} T_{ser,\mathcal{T}_{i+1}-1}(k) \leq 2 \sum_{k=1}^{K} T_{ser,\mathcal{T}_i}(k)$. We can then rewrite equation (17) as

$$M + 1/\gamma \geq \sum_{m=1}^{M} \mathcal{N}_m = L.$$

Due to one communication including one upload and one download, the communication cost in one episode is at most $2(M + 1/\gamma)$ (following the definition of (2)). We can then bound the total communication cost

$$\mathcal{C}(\tau) \leq 2\Big( (M + 1/\gamma) \log_2(\tau) \Big). \tag{18}$$

In the light of (18), setting of the Theorem 1 and (22) (the upper bound of the sample complexity $\tau$), we can bound the communication cost

$$\mathcal{C}(\tau) = \tilde{O}(KM). \tag{19}$$

$\square$

## B.2 UPPER BOUND SAMPLE COMPLEXITY $\tau$

Combining the breaking condition in Algorithm 1 (line 14~16), and the definition of $B(\tau)$, we have

$$\epsilon \geq B(\tau) = \hat{\Delta}_\tau^{ser}(j_\tau^{ser}, i_\tau^{ser}) + \beta_\tau^{ser}(i_\tau^{ser}, j_\tau^{ser}).$$

Let's first consider the case when the empirically best arm on the server side is not the optimal arm, i.e., $i_{ser,\tau} \neq k^*$. By the definition of $j_{ser,\tau}$, we have

$$\hat{\Delta}_{ser,\tau}(j_{ser,\tau}, i_{ser,\tau}) + \beta_{ser,\tau}(i_{ser,\tau}, j_{ser,\tau}) \geq \hat{\Delta}_{ser,\tau}(k^*, i_{ser,\tau}) + \alpha_{ser,\tau}^{M}(i_{ser,\tau}, k^*).$$

Recall that $\hat{k}^* = i_{ser,\tau}$ is the empirically best arm. Therefore, we have

$$\epsilon \geq \hat{\Delta}_{ser,\tau}(k^*, \hat{k}^*) + \alpha_{ser,\tau}^{M}(\hat{k}^*, k^*) \geq \Delta(k^*, \hat{k}^*),$$

where the second inequality is due to Lemma 2 below (proof of Lemma 2 is at the end of this section).

**Lemma 2.** *Following the setting of Theorem 1, we can establish the exploration bonuses*

$$\alpha_{m_t,t}^M(k) = \sigma \sqrt{\frac{2}{T_{m_t,t}(k)} \log\left(\frac{4K}{\delta}\left((1+\gamma M)\sum_{k=1}^{K} T_{m_t,t}(k)\right)^2\right)}$$

$$\alpha_{ser,t}^M(k) = \sigma \sqrt{\frac{2}{T_{ser,t}(k)} \log\left(\frac{4K}{\delta}\left((1+\gamma M)\sum_{k=1}^{K} T_{ser,t}(k)\right)^2\right)},$$

*for all $t \in [K+1, \tau]$, $k \in \mathcal{A}$, and the event*

$$\mathcal{I} = \left\{\forall k \in \mathcal{A}, \forall t \in [K+1, \tau], |\hat{\mu}_{m_t,t}(k) - \mu(k)| \le \beta_{m_t,t}(k), |\hat{\mu}_{ser,t}(k) - \mu(k)| \le \alpha_{ser,t}^M\right\}.$$

*We have $\mathcal{P}(\mathcal{I}) \ge 1 - \delta$.*

Furthermore, if $i_{ser,t} = k^*$, we have $\epsilon \ge \Delta(k^*, \hat{k}^*) = 0$. The discussion above implies $\hat{k}^*$ output by FAMABPE satisfies the $(\epsilon, \delta)$-condition (1).

We now continue to bound the sample complexity $\tau$. First, we need to establish Lemma 3 below, which upper bounds $T_{ser,\tau}(k)$, the number of observations on arm $k$ that is available to the server at $\tau$.

**Lemma 3.** *Under the setting of Theorem 1, if event $\mathcal{I}$ happens, we can bound*

$$T_{ser,\tau}(k) \le \frac{2\sigma^2 \log\left(4K\left((1+M\gamma)\sum_{s=1}^{K} T_{ser,\tau}(s)\right)^2/\delta\right)}{\max\left(\frac{\Delta(k^*,k)+\epsilon}{3}, \epsilon\right)^2} + \gamma M \sum_{s=1}^{K} T_{ser,\tau}(s), \forall k \in \mathcal{A}.$$

With Lemma 3, we have

$$\sum_{k=1}^{K} T_{ser,\tau}(k) \le \sum_{k=1}^{K} \frac{2\sigma^2 \log\left(4K\left((1+\gamma M)\sum_{s=1}^{K} T_{ser,\tau}(s)\right)^2/\delta\right)}{\max\left(\frac{\Delta(k^*,k)+\epsilon}{3}, \epsilon\right)^2} + \gamma KM \sum_{s=1}^{K} T_{ser,\tau}(s)$$

$$\le \frac{1}{1-\gamma KM} \sum_{k=1}^{K} \frac{2\sigma^2 \log\left(4K\left((1+\gamma M)\sum_{s=1}^{K} T_{ser,\tau}(s)\right)^2/\delta\right)}{\max\left(\frac{\Delta(k^*,k)+\epsilon}{3}, \epsilon\right)^2}$$

$$\le \frac{1}{1-\gamma KM} \sum_{k=1}^{K} \frac{2\sigma^2 \log\left(4K\left((1+\gamma M)\tau\right)^2/\delta\right)}{\max\left(\frac{\Delta(k^*,k)+\epsilon}{3}, \epsilon\right)^2}.$$

The last inequality is due to $\sum_{k=1}^{K} T_{ser,t}(k) \le \tau$ (Remark 3). Based on the relation between $\tau$, $\sum_{m=1}^{M} \sum_{k=1}^{K} T_{m,\tau}^{loc}(k)$, and $\sum_{k=1}^{K} T_{ser,\tau}(k)$ (Remark 3), we have

$$\tau = \sum_{k=1}^{K} T_{\tau}^{all}(k) = \sum_{k=1}^{K} T_{ser,\tau}(k) + \sum_{m=1}^{M} \sum_{k=1}^{K} T_{m,\tau}^{loc}(k)$$

$$\le (1+\gamma M) \sum_{k=1}^{K} T_{ser,\tau}(k)$$

$$\le \frac{1+\gamma M}{1-\gamma KM} \sum_{k=1}^{K} \frac{2\sigma^2 \log\left(4K\left((1+\gamma M)\tau\right)^2/\delta\right)}{\max\left(\frac{\Delta(k^*,k)+\epsilon}{3}, \epsilon\right)^2} \tag{20}$$

$$= \frac{1+\gamma M}{1-\gamma KM} H_{\epsilon}^M 2 \log\left(\frac{4K\left((1+\gamma M)\tau\right)^2}{\delta}\right),$$

where the first inequality is due to Lemma 4, the second inequality is due to the inequality we established above, and the third is by definition of $H_\epsilon^M$.

Recalling that in Theorem 1 we suppose $\gamma = 1/(2MK)$. Let $\tau'$ be a parameter that satisfies

$$\tau' \le \tau = \frac{M + 1/(2K)}{M - 1/2} H_\epsilon^M 2 \log \left( \frac{4K\left((1 + 1/(2K))\tau'\right)^2}{\delta} \right),$$

where the equality is owing to the definition of the $\gamma$ and (20). Due to the fact that $\sqrt{x} \ge \log(x)$ holds when $x > 0$, we have

$$\begin{aligned}
\tau' &\le \frac{M + 1/(2K)}{M - 1/2} H_\epsilon^M 2 \log \left( \left( \frac{4K(1 + 1/(2K))\tau'}{\delta^{1/2}} \right)^2 \right) \\
&\le \frac{M + 1/(2K)}{M - 1/2} H_\epsilon^M 4 \sqrt{\frac{4K(1 + 1/(2K))\tau'}{\delta^{1/2}}} \\
&\le \left( \frac{M + 1/(2K)}{M - 1/2} H_\epsilon^M 4 \right)^2 \frac{4K(1 + 1/(2K))}{\delta^{1/2}}.
\end{aligned} \tag{21}$$

We define the last term of (21) as $\Lambda$. Then based on (20) and (21), we can finally get

$$\tau \le \frac{M + 1/(2K)}{M - 1/2} H_\epsilon^M 2 \log \left( \frac{4K\left((1 + 1/(2K))\Lambda\right)^2}{\delta} \right), \tag{22}$$

where

$$\Lambda = \left( \frac{M + 1/(2K)}{M - 1/2} H_\epsilon^M 4 \right)^2 \frac{4K(1 + 1/(2K))}{\delta^{1/2}}.$$

**Lemma 4.** *Following the setting of Theorem 1, we have $\forall t \in [K + 1, \tau]$*

$$\sum_{k=1}^K T_{ser,t}(k) > 1/\gamma \sum_{k=1}^K T_{m_t,t}^{loc}(k).$$

**Proof for Lemma 2, Lemma 3, and Lemma 4**   In the following paragraphs, we provide the detailed proof for the lemmas used above.

*Proof of Lemma 4.*   For every round $t \in [K + 1, \tau]$, if the agent $m_t$ communicates with the server at round $t$, we have

$$\sum_{k=1}^K T_{ser,t}(k) \ge 1/\gamma \sum_{k=1}^K T_{m_t,t}^{loc}(k) = 0.$$

The inequality holds by line 19 of Algorithm 1.

Else, according to the triggered condition of Algorithm 1, if agent $m_t$ does not communicate with the server in round $t$, we have

$$\sum_{k=1}^K (T_{m_t,t}(k) + T_{m_t,t}^{loc}(k)) \le (1 + \gamma) \sum_{k=1}^K T_{m_t,t}(k)$$

$$\sum_{k=1}^K T_{m_t,t}^{loc}(k) \le \gamma \sum_{k=1}^K T_{m_t,t}(k).$$

With the fact that $\sum_{k=1}^K T_{m_t,t}(k) \le \sum_{k=1}^K T_{ser,t}(k)$ (Remark 3), we can finally get

$$\sum_{k=1}^K T_{m_t,t}^{loc}(k) \le \gamma \sum_{k=1}^K T_{ser,t}(k).$$

Here we finish the proof of Lemma 4. □

Based on Lemma 4, we can prove Lemma 2 as shown below.

*Proof of Lemma 2.* Due to $\hat{\mu}_{m_t,t}(k)$ and $T_{m_t,t}(k)$, $\forall k \in \mathcal{A}$, $t \in [K+1, \tau]$ are all downloaded from the server and they would remain unchanged until the next round agent $m_t$ communicates with the server. This implies $\forall t_1 \in [K+1, \tau]$, there exists a $t_2 \in [K+1, \tau]$ which satisfies

$$\alpha^M_{m_{t_1},t_1}(k) = \alpha^M_{ser,t_2}(k) \text{ and } \hat{\mu}_{m_{t_1},t_1}(k) = \hat{\mu}_{ser,t_2}(k), \ \forall k \in \mathcal{A}.$$

Hence, we can derive

$$\mathcal{P}(\mathcal{I}) = \mathcal{P}\left( \forall k \in \mathcal{A}, \forall t \in [K+1, \tau], \ |\hat{\mu}_{ser,t}(k) - \mu(k)| \leq \alpha^M_{ser,t}(k) \right). \tag{23}$$

We define $\mathcal{I}^c$ as the contradicted event of $\mathcal{I}$. Utilizing the union bound, it can be decomposed by

$$\mathcal{P}(\mathcal{I}^c) \leq \sum_{k=1}^{K} \sum_{t=K+1}^{\tau} \mathcal{P}\left( |\hat{\mu}_{ser,t}(k) - \mu(k)| > \alpha^M_{ser,t}(k) \right). \tag{24}$$

With the help of the Hoeffeding inequality (Lemma 13), it has

$$\begin{aligned}
\mathcal{P}\left( |\hat{\mu}_{ser,t}(k) - \mu(k)| > \alpha^M_{ser,t}(k) \right) &\leq e^{-\alpha^M_{ser,t}(k)^2 T_{ser,t}(k)/2\sigma^2} \\
&= e^{-\log\left( \frac{4K}{\delta} \left( (1+\gamma M) \sum_{k=1}^{K} T_{ser,t}(k) \right)^2 \right)} \\
&= \frac{\delta}{4K\left( (1+\gamma M) \sum_{k=1}^{K} T_{ser,t}(k) \right)^2} \\
&\leq \frac{\delta}{4Kt^2}.
\end{aligned} \tag{25}$$

The first equality is owing to the definition of $\alpha_{ser,t}(k)$ and the last inequality is owing to $t = \sum_{k=1}^{K} T_{ser,t}(k) + \sum_{m=1}^{M} \sum_{k=1}^{K} T^{loc}_{m,t}(k) \leq (1+\gamma M) \sum_{k=1}^{K} T_{ser,t}(k)$ (Lemma 4). Substituting the last term of (25) into (24), we can finally bound

$$\sum_{k=1}^{K} \sum_{t=K+1}^{\tau} \mathcal{P}\left( |\hat{\mu}^{ser}_t(k) - \mu(k)| > \alpha^M_{ser,t}(k) \right) \leq \delta$$

and $\mathcal{P}(\mathcal{I}) = 1 - \mathcal{P}(\mathcal{I}^c) \geq 1 - \delta$. Here we finish the proof of Lemma 2. $\qquad\square$

Before proving Lemma 3, we first need to establish Lemma 5 below.

**Lemma 5.** *We define $k_{ser,t}$ in round $t \in [K+1, \tau]$ as $k_{ser,t} = \arg\max_{k \in \{i_{ser,t}, j_{ser,t}\}} \alpha^M_{ser,t}(k)$. Following the setting of Theorem 1, if event $\mathcal{I}$ happens, we can bound the index $B(t)$ as*

$$B(t) \leq \min\left( 0, -\Delta(k^*, k^{ser}_t) + 4\beta_{ser,t}(k_{ser,t}) \right) + 2\alpha^M_{ser,t}(k_{ser,t}). \tag{26}$$

*Proof of Lemma 5.* This proof follows the idea of Gabillon et al. [2012]. Consider the case when $i_{ser,t} = k^*$, we have

$$\begin{aligned}
B(t) &= \hat{\Delta}_{ser,t}(j_{ser,t}, i_{ser,t}) + \alpha^M_{ser,t}(i_{ser,t}, j_{ser,t}) \\
&\leq \hat{\Delta}_{ser,t}(j_{ser,t}, i_{ser,t}) + 2\alpha^M_{ser,t}(k_{ser,t}) \\
&\leq \Delta(j_{ser,t}, i_{ser,t}) + 4\alpha^M_{ser,t}(k_{ser,t}) \\
&= -\Delta(k^*, j_{ser,t}) + 4\alpha^M_{ser,t}(k_{ser,t}) \\
&\leq -\Delta(k^*, k_{ser,t}) + 4\alpha^M_{ser,t}(k_{ser,t}),
\end{aligned} \tag{27}$$

where the first inequality is owing to the definition of $k_{ser,t}$ and the second inequality is owing to the definition of the event $\mathcal{I}$. Then, consider the case when $j_{ser,t} = k^*$, we have

$$
\begin{aligned}
B(t) &= \hat{\Delta}_{ser,t}(j_{ser,t}, i_{ser,t}) + \alpha_{ser,t}^M(i_{ser,t}, j_{ser,t}) \\
&\leq \hat{\Delta}_{ser,t}(j_{ser,t}, i_{ser,t}) + 2\alpha_{ser,t}^M(k_{ser,t}) \\
&\leq -\hat{\Delta}_{ser,t}(j_{ser,t}, i_{ser,t}) + 2\alpha_{ser,t}^M(k_{ser,t}) \\
&\leq -\Delta(j_{ser,t}, i_{ser,t}) + 4\alpha_{ser,t}^M(k_{ser,t}) \\
&= -\Delta(k^*, i_{ser,t}) + 4\alpha_{ser,t}^M(k_{ser,t}) \\
&\leq -\Delta(k^*, k_{ser,t}) + 4\alpha_{ser,t}^M(k_{ser,t}),
\end{aligned}
\tag{28}
$$

where the second inequality is owing to the definition of the $i_{ser,t}$ (line 13 of Algorithm 1).

Combine (27) and (28), it yields

$$
B(t) \leq -\Delta(k^*, k_{ser,t}) + 4\alpha_{ser,t}^M(k_{ser,t}),
\tag{29}
$$

when $i_{ser,t} = k^*$ or $j_{ser,t} = k^*$. Furthermore, due to $B(t) = \hat{\Delta}_{ser,t}(j_{ser,t}, i_{ser,t}) + \alpha_{ser,t}^M(j_{ser,t}, i_{ser,t})$ and $\hat{\Delta}_{ser,t}(j_{ser,t}, i_{ser,t}) \leq 0$, we can derive $B(t) \leq \alpha_{ser,t}^M(j_{ser,t}, i_{ser,t}) \leq 2\alpha_{ser,t}^M(k_{ser,t})$. In the light of (29), we can finally get

$$
B(t) \leq \min\left(0, \Delta(k^*, k_{ser,t}) + 2\alpha_{ser,t}^M(k_{ser,t})\right) + 2\alpha_{ser,t}^M(k_{ser,t}).
\tag{30}
$$

We further consider the case when $i_{ser,t} \neq k^*$ and $j_{ser,t} \neq k^*$, then we can derive

$$
\begin{aligned}
B(t) &= \hat{\Delta}_{ser,t}(j_{ser,t}, i_{ser,t}) + \alpha_{ser,t}^M(i_{ser,t}, j_{ser,t}) \\
&\leq \Delta(j_{ser,t}, i_{ser,t}) + 2\alpha_{ser,t}^M(i_{ser,t}, j_{ser,t}) \\
&= \Delta(j_{ser,t}, k^*) + \Delta(k^*, i_{ser,t}) + 2\alpha_{ser,t}^M(i_{ser,t}, j_{ser,t}) \\
&\leq \Delta(j_{ser,t}, k^*) + 3\alpha_{ser,t}^M(i_{ser,t}, j_{ser,t}) \\
&\leq \Delta(j_{ser,t}, k^*) + 6\alpha_{ser,t}^M(k_{ser,t}) \\
&= -\Delta(k^*, j_{ser,t}) + 6\alpha_{ser,t}^M(k_{ser,t}),
\end{aligned}
\tag{31}
$$

where the second equality is owing to $\Delta(i_{ser,t}, j_{ser,t}) = \Delta(i_{ser,t}, k^*) + \Delta(k^*, j_{ser,t})$ and second inequality is owing to

$$
\begin{aligned}
\alpha_{ser,t}^M(i_{ser,t}, j_{ser,t}) &\geq \hat{\Delta}_{ser,t}(j_{ser,t}, i_{ser,t}) + \alpha_{ser,t}^M(i_{ser,t}, j_{ser,t}) \\
&\geq \hat{\Delta}_{ser,t}(k^*, i_{ser,t}) + \alpha_{ser,t}^M(i_{ser,t}, k^*) \\
&\geq \Delta(k^*, i_{ser,t}).
\end{aligned}
$$

Similar to (31), we also can show

$$
\begin{aligned}
B(t) &= \hat{\Delta}_{ser,t}(j_{ser,t}, i_{ser,t}) + \alpha_{ser,t}^M(i_{ser,t}, j_{ser,t}) \\
&\leq \alpha_{ser,t}^M(i_{ser,t}, j_{ser,t}) \\
&\leq -\Delta(k^*, i_{ser,t}) + \hat{\Delta}_{ser,t}(k^*, i_{ser,t}) + \alpha_{ser,t}^M(k^*, i_{ser,t}) + \alpha_{ser,t}^M(i_{ser,t}, j_{ser,t}) \\
&\leq -\Delta(k^*, i_{ser,t}) + \hat{\Delta}_{ser,t}(j_{ser,t}, i_{ser,t}) + \alpha_{ser,t}^M(j_{ser,t}, i_{ser,t}) + \alpha_{ser,t}^M(i_{ser,t}, j_{ser,t}) \\
&\leq -\Delta(k^*, i_{sert}) + 4\alpha_{ser,t}^M(k_{ser,t}).
\end{aligned}
\tag{32}
$$

The second inequality is due to the definition of the even $\mathcal{I}$ and the third inequality is due to the definition of the $j_{ser,t}$.

Combine (31) and (32), it yields

$$
B(t) \leq -\Delta(k^*, k_{ser,t}) + 6\alpha_{ser,t}^M(k_{ser,t})
\tag{33}
$$

when $i_{ser,t} \neq k^*$ and $j_{ser,t} \neq k^*$. In the light of (33), we can finally get

$$
B(t) \leq \min\left(0, -\Delta(k^*, k_{ser,t}) + 4\alpha_{ser,t}^M(k_{ser,t})\right) + 2\alpha_{ser,t}^M(k_{ser,t}).
\tag{34}
$$

Combine (30) and (34), then we can finish the proof of Lemma 2. $\square$

*Proof of Lemma 3.* Suppose agent $m$ communicates in round $t_1$ and $t_2$, then from round $t \in [t_1 + 1, t_2]$, owing to $\hat{\mu}_{m,t}(k)$ and $T_{m,t}(k)$ remain unchanged, $k_{m,t}$ would not change either (we highlight this knowledge in Remark **??**). We define at round $t_k \neq \tau$, an agent $m$ communicates with the server and $k_{ser,t_k} = k$, $\forall k \in \mathcal{A}$ (the definition of the $k_{ser,t}$ is provided in Lemma 5). And after round $t_k$, when any agent communicating with the server, $k_{ser,t} \neq k$. This implies

$$
\begin{aligned}
T_{ser,\tau}(k) \leq & T_{m,t_k+1}(k) + (\gamma M) \sum_{s=1}^{K} T_{ser,\tau}(s) \\
= & T_{ser,t_k}(k) + (\gamma M) \sum_{s=1}^{K} T_{ser,\tau}(s).
\end{aligned}
\tag{35}
$$

The inequality holds due to for $t \in [t_k + 1, \tau]$, $\forall m \in \mathcal{M}$ would upload $T_{m,t}^{loc}(k) > 0$ to the server at most one time (according to the definition of $t_k$) and $T_{m,t}^{loc}(k) \leq \sum_{s=1}^{K} T_{m,t}^{loc}(s) \leq \gamma \sum_{s=1}^{K} T_{ser,\tau}(s)$ according to $t \leq \tau$ and the Lemma 4.

We would further bound $T_{ser,t_k}(k) = T_{m,t_k+1}(k)$. According to agent $m$ sets $k_{ser,t_k} = k$ and the definition of the breaking condition (line 14~16 in Algorithm 1), with Lemma 5, we can derive

$$
\begin{aligned}
\epsilon \leq & B(t_k) \\
\leq & \min\left(0, -\Delta(k^*, k) + 4\alpha_{ser,t_k}^{M}(k)\right) + 2\alpha_{ser,t_k}^{M}(k).
\end{aligned}
\tag{36}
$$

Substituting (6) (the definition of $\alpha_{ser,t_k}^{M}(k)$) into (36), we have

$$
T_{ser,t_k}(k) \leq \frac{2\sigma^2 \log\left(4K\left((1+\gamma M)\sum_{s=1}^{K} T_{ser,t_k}(s)\right)^2/\delta\right)}{\max\left(\frac{\Delta(k^*,k)+\epsilon}{3}, \epsilon\right)^2}.
$$

With (35) and $t_k < \tau$, we can finally bound $T_{ser,\tau}(k)$, i.e.

$$
T_{ser,\tau}(k) \leq \frac{2\sigma^2 \log\left(4K\left((1+\gamma M)\sum_{s=1}^{K} T_{ser,\tau}(s)\right)^2/\delta\right)}{\max\left(\frac{\Delta(k^*,k)+\epsilon}{3}, \epsilon\right)^2} + \gamma M \sum_{s=1}^{K} T_{ser,\tau}(s).
$$

Here we finish the proof of Lemma 3. $\qquad\square$

## C  PROOF OF THEOREM 2

Recall that the upper confidence bounds of the agent $m$ and server are defined as $\alpha_{m,t}(i,j) = \|\mathbf{y}(i,j)\|_{\mathbf{V}_{m_t,t}^{-1}} C_{m,t}$ and $\alpha_{ser,t}^{L}(i,j) = \|\mathbf{y}(i,j)\|_{\mathbf{V}_{ser,t}^{-1}} C_{ser,t}$, respectively. The estimated model parameters of the agent $m$ and server are denoted as $\hat{\boldsymbol{\theta}}_{m,t}$ and $\hat{\boldsymbol{\theta}}_{ser,t}$, respectively. Besides, we also provide the Remark 4 to specifically illustrate the relationship between some most important data used in the proof of Theorem 2.

**Remark 4** (Global and local data of the linear case). *Due to the transmitted data in the linear case being more complicated than the data in MAB. We here provide new notations to clarify the relations between global data and local data. The following matrix and vector denote the global data*

$$
\begin{aligned}
\mathbf{V}_t^{all} &= \lambda \mathbf{I} + \sum_{s=1}^{t} \mathbf{x}_{m_s,s} \mathbf{x}_{m_s,s}^{\top} = \mathbf{V}_{ser,t} + \sum_{m=1}^{M} \mathbf{V}_{m,t}^{loc}, \\
\mathbf{b}_t^{all} &= \sum_{s=1}^{t} \mathbf{x}_{m_s,s} r_{m_s,s} = \mathbf{b}_{ser,t} + \sum_{m=1}^{M} \mathbf{b}_{m,t}^{loc}, \\
T_t^{all}(k) &= \sum_{s=1}^{t} \mathbb{1}\{k = k_{m_s,s}\} = T_{ser,t}(k) + \sum_{m=1}^{M} T_{m,t}^{loc}(k), \ \forall k \in \mathcal{A}.
\end{aligned}
$$

*We define $N_{m,t}$ as the final round when agent $m$ communicates with the server at the end of round $t$. The collected data of agent $m$ that has not been uploaded to the server is provided as follows*

$$\mathbf{V}_{m,t}^{loc} = \sum_{s=N_{m,t}+1}^{t} \mathbb{1}\{m_s = m\}\mathbf{x}_{m_s,s}\mathbf{x}_{m_s,s}^{\top},$$

$$\mathbf{b}_{m,t}^{loc} = \sum_{s=N_{m,t}+1}^{t} \mathbb{1}\{m_s = m\}\mathbf{x}_{m_s,s}r_{m_s,s}$$

$$T_{m,t}^{loc}(k) = \sum_{s=N_{m,t}+1}^{t} \mathbb{1}\{m_s = m, k = k_{m_s,s}\}, \ \forall k \in \mathcal{A}.$$

*Similarly, the data that has been uploaded to the server yields*

$$\mathbf{V}_{ser,t} = \lambda\mathbf{I} + \sum_{m=1}^{M}\sum_{s=1}^{N_{m,t}} \mathbb{1}\{m_s = m\}\mathbf{x}_{m_s,s}\mathbf{x}_{m_s,s}^{\top},$$

$$\mathbf{b}_{ser,t} = \sum_{m=1}^{M}\sum_{s=1}^{N_{m,t}} \mathbb{1}\{m_s = m\}\mathbf{x}_{m_s,s}r_{m_s,s}$$

$$T_{ser,t}(k) = \sum_{m=1}^{M}\sum_{s=1}^{N_{m,t}} \mathbb{1}\{m_s = m, k = k_{m_s,s}\}, \ \forall k \in \mathcal{A}.$$

*According to the communication protocol, the local data of every agent $m \in \mathcal{M}$ can be represented by*

$$\mathbf{V}_{m,t} = \mathbf{V}_{ser,N_{m,t}}, \ \mathbf{b}_{m,t} = \mathbf{b}_{ser,N_{m,t}}, \ T_{m,t}(k) = T_{ser,N_{m,t}}(k), \ \forall k \in \mathcal{A}.$$

*Accordingly, we have $V_{m,t} \preceq V_{ser,t}$ and $\sum_{k=1}^{K} T_{m,t}(k) \leq \sum_{k=1}^{K} T_{ser,t}(k), \forall t \in [K+1, \tau]$.*

**Proof sketch of Theorem 2**    The proof of Theorem 2 also consists of three main components, i.e., a) the sample complexity $\tau$; b) the communication cost $\mathcal{C}(\tau)$; c) the estimated best arm satisfies Eq (1). Specifically, to upper bound the total communication cost $C(\tau)$, we utilize the property that the agents communicating with the server when at least one of the two events (line 11 of Algorithm 2) is triggered (Lemma 6). To upper bound the sample complexity $\tau$, we first establish the relationship between $\sum_{k=1}^{K} T_{ser,t}(k)$ and $\sum_{k=1}^{K} T_{m,t}^{loc}(k)$ and the relationship between $\mathbf{V}_{ser,t}$ and $\mathbf{V}_{m,t}^{loc}$ based on the hybrid event triggered strategy (Lemma 9). Then, we design the exploration bonuses by $\sum_{k=1}^{K} T_{m_t,t}(k)$, $\mathbf{V}_{m_t,t}$, $\sum_{k=1}^{K} T_{ser,t}(k)$ and $\mathbf{V}_{ser,t}$ (Lemma 7). Furthermore, we bound the matrix norms $\|\mathbf{y}(i,j)\|_{\mathbf{V}_{m_t,t}^{-1}}$ and $\|\mathbf{y}(i,j)\|_{\mathbf{V}_{ser,t}^{-1}}$, $\forall i, j \in \mathcal{A}$ based on the arm selection strategy (Lemma 10). Combine these knowledge, we can bound $T_{ser,\tau}(k), \forall k \in \mathcal{A}$ (Lemma 11 and 8). Finally, utilizing the knowledge of Remark 4, we can bound $T_{\tau}^{all}(k), \forall k \in \mathcal{A}$, and $\tau = \sum_{k=1}^{K} T_{\tau}^{all}(k)$. Similar to the MAB setting, the guarantee on finding the best arm directly follows the property of the breaking index, i.e., if $B(\tau) \leq \epsilon$, then $\Delta(k^*, \hat{k}^*) \leq \epsilon$ with probability at least $1 - \delta$.

## C.1   UPPER BOUND COMMUNICATION COST $\mathcal{C}(\tau)$

**Lemma 6** (Communication cost of the hybrid event-triggered strategy in Algorithm 2)**.** *Under the setting of Theorem 2, the total triggered number of the first event (line 11 of Algorithm 2) can be bounded by*

$$(M + 1/\gamma_1)d\log_2\left(1 + \frac{\tau}{\lambda d}\right) \tag{37}$$

*and the total triggered number of the second event can be bounded by*

$$(M + 1/\gamma_2)\log_2(\tau).$$

*The total communication cost can be bounded by*

$$\mathcal{C}(\tau) \leq 2\left((M + 1/\gamma_1)d\log_2\left(1 + \frac{\tau}{\lambda d}\right) + (M + 1/\gamma_2)\log_2(\tau)\right).$$

*Proof of Lemma 6.* The triggered number of the second event can be bounded by Lemma 1. Besides, we can bound the triggered number of the first event similar to He et al. [2022]. The proof of (37) also can be divided into two sections, in the first section, we would divide the sample complexity $\tau$ into $\log_2(1 + \tau/\lambda d)$ episodes, then we would analysis the upper bound of the triggered number of the first event in each episode. We define

$$\mathcal{T}_i = \min \left\{ t \in |\tau|, \ \det(\mathbf{V}_{ser,t}) \geq 2^i \lambda^d \right\}$$

and the set of all rounds into episodes $\{\mathcal{T}_i, \mathcal{T}_i + 1, ..., \mathcal{T}_{i+1} - 1\}, \forall i \geq 0$. By the Lemma 14, we can bound

$$\det(\mathbf{V}_{ser,\tau}) \leq \lambda^d \left( 1 + \frac{\tau}{\lambda d} \right)^d.$$

Accordingly, the number of the episode can be bounded by

$$\max\{i \geq 0\} = \log_2 \left( \frac{\det(\mathbf{V}_{ser,\tau})}{\lambda^d} \right) \leq d \log_2 \left( 1 + \frac{\tau}{\lambda d} \right).$$

We then prove $\forall i \geq 0$, from round $\mathcal{T}_i$ to $\mathcal{T}_{i+1} - 1$, the triggered number of the first event can be bounded by $M + 1/\gamma_1$. We first define the number of agents $m$ triggers the first event in $\mathcal{T}_i$ to $\mathcal{T}_{i+1} - 1$ as $\mathcal{N}_m$, the sequence of agent $m$ triggers the first event in round $\mathcal{T}_i$ to $\mathcal{T}_{i+1} - 1$ as $t_1^m, ..., t_{\mathcal{N}_m}^m$, the number of every agent triggers the first event in $\mathcal{T}_i$ to $\mathcal{T}_{i+1} - 1$ as $L$ and the sequence of the first event be triggered in $\mathcal{T}_i$ to $\mathcal{T}_{i+1} - 1$ as $t_{i,1}, ..., t_{i,L}$. According to the definition of the first event, we have

$$\det(\mathbf{V}_{m_t,t} + \mathbf{V}_{m_t,t}^{loc}) > (1 + \gamma_1)\det(\mathbf{V}_{m_t,t}).$$

Then, $\forall m \in \mathcal{M}, \ j \in |\mathcal{N}_m|/\{1\}$, we have

$$\det(\mathbf{V}_{ser,\mathcal{T}_i} + \mathbf{V}_{m,t_j^m}^{loc}) \geq \frac{\det(\mathbf{V}_{ser,\mathcal{T}_i})}{\det(\mathbf{V}_{m,t_j^m})}\det(\mathbf{V}_{m,t_j^m} + \mathbf{V}_{m,t_j^m}^{loc}) \geq (1 + \gamma_1)\det(\mathbf{V}_{ser,\mathcal{T}_i}) \tag{38}$$

The inequality holds due to $\forall j \in |\mathcal{N}_m|/\{1\}$, $\mathbf{V}_{ser,\mathcal{T}_i} \preceq \mathbf{V}_{m,t_j^m}$ and Lemma 15. The above inequality implies $\forall t_{i,l} \geq t_2^{m_{t_{i,l}}}$

$$\begin{aligned}
\det(\mathbf{V}_{ser,t_{i,l}} - \mathbf{V}_{ser,t_{i,l-1}}) &= \det(\mathbf{V}_{ser,t_{i,l-1}} + \mathbf{V}_{m_{t_{i,l}},t_{i,l}}^{loc}) - \det(\mathbf{V}_{ser,t_{i,l-1}}) \\
&\geq \det(\mathbf{V}_{ser,\mathcal{T}_i} + \mathbf{V}_{m_{t_{i,l}},t_{i,l}}^{loc}) - \det(\mathbf{V}_{ser,\mathcal{T}_i}) \\
&\geq \gamma_1 \det(\mathbf{V}_{ser,\mathcal{T}_i}).
\end{aligned}$$

The first inequality holds is owing to Lemma 16 and the last inequality holds is owing to (38).

Finally we can bound $L = \sum_{m=1}^M \mathcal{N}_m$

$$\begin{aligned}
\det(\mathbf{V}_{ser,\mathcal{T}_{i+1}-1}) - \det(\mathbf{V}_{ser,\mathcal{T}_i}) &= \sum_{l=1}^{L-1} \left( \det(\mathbf{V}_{ser,t_{i,l+1}}) - \det(\mathbf{V}_{ser,t_{i,l}}) \right) \\
&\geq \gamma_1 \sum_{m=1}^M (\mathcal{N}_m - 1)\det(\mathbf{V}_{ser,\mathcal{T}_i}).
\end{aligned} \tag{39}$$

Due to the definition of the episode, it has $2\det(\mathbf{V}_{ser,\mathcal{T}_i}) \geq \det(\mathbf{V}_{ser,\mathcal{T}_{i+1}-1})$. We can rewrite equation (39) as

$$M + 1/\gamma_1 \geq \sum_{m=1}^M \mathcal{N}_m.$$

We can then bound the total triggered number of the first event by

$$(M + 1/\gamma_2) \log_2 \left( 1 + \frac{\tau}{\lambda d} \right).$$

Due to the communication would happen when at least one of the events is triggered, the total communication round is smaller or equal to the triggered number of two events. Hence, the total communication number from $t = K + 1$ to $t = \tau$ can be bounded by

$$(M + 1/\gamma_1)d\log_2\left(1 + \frac{\tau}{\lambda d} + \tau^{1/d}\right).$$

Furthermore, due to one communication includes one upload and one download, the total communication cost can be bounded by

$$\mathcal{C}(\tau) \leq 2\left((M + 1/\gamma_1)d\log_2\left(1 + \frac{\tau}{\lambda d}\right) + (M + 1/\gamma_2)\log_2(\tau)\right). \tag{40}$$

With (40), (44) (the upper bound of the sample complexity), and the setting of Theorem (2), we can bound the communication cost

$$\mathcal{C}(\tau) = O\left(\max(2MK, M^2)\right).$$

Here we finish the proof of Lemma 6. $\qquad\square$

## C.2 UPPER BOUND SAMPLE COMPLEXITY $\tau$

Combine the breaking condition of the Algorithm 2 (line 14~16) and definition of $B(\tau)$, we have

$$\epsilon \geq \hat{\Delta}_{ser,\tau}(j_{ser,\tau}, i_{ser,\tau}) + \alpha^L_{ser,\tau}(i_{ser,\tau}, j_{ser,\tau}) = B(\tau).$$

Let's first consider the case when the empirically best arm on the server side is not the optimal arm, i.e., $i_{ser,\tau} \neq k^*$. By the definition of the $j_{ser,t}$, we have

$$\hat{\Delta}_{ser,\tau}(j_{ser,\tau}, i_{ser,\tau}) + \alpha^L_{ser,\tau}(i_{ser,\tau}, j_{ser,\tau}) \geq \hat{\Delta}_{ser,\tau}(k^*, i_{ser,\tau}) + \alpha^L_{ser,\tau}(i_{ser,\tau}, k^*).$$

Recall that $\hat{k}^* = i_{ser,\tau}$ is the estimated best arm. Therefore, we have

$$\epsilon \geq \hat{\Delta}_{ser,\tau}(k^*, \hat{k}^*) + \alpha^L_{ser,\tau}(\hat{k}^*, k^*) \geq \Delta(k^*, \hat{k}^*),$$

where the second inequality is due to Lemma 7 below (proof of Lemma 7 is at the end of the section).

**Lemma 7.** *Following the setting of Theorem 2, we define event*

$$\mathcal{I} = \left\{\forall i, j \in \mathcal{A}, \forall t \in [K + 1, \tau], |\hat{\Delta}_{m_t,t}(i,j) - \Delta(i,j)| \leq \alpha^L_{m_t,t}(i,j), |\hat{\Delta}_{ser,t}(i,j) - \Delta(i,j)| \leq \alpha^L_{ser,t}(i,j)\right\}$$

*where*

$$\alpha^L_{m_t,t}(i,j) = \left(\sqrt{\lambda} + \left(\sqrt{2\gamma_1}M + \sqrt{1 + \gamma_1 M}\right)\left(\sigma\sqrt{d\log\left(\frac{2}{\delta}\left(1 + \frac{(1 + \gamma_2 M)\sum_{k=1}^K T_{m_t,t}(k)}{\min(\gamma_1, 1)\lambda}\right)\right)}\right)\right)\|\mathbf{y}(i,j)\|_{\mathbf{V}^{-1}_{m_t,t}}$$

$$\alpha^L_{ser,t}(i,j) = \left(\sqrt{\lambda} + \left(\sqrt{2\gamma_1}M + \sqrt{1 + \gamma_1 M}\right)\left(\sigma\sqrt{d\log\left(\frac{2}{\delta}\left(1 + \frac{(1 + \gamma_2 M)\sum_{k=1}^K T_{ser,t}(k)}{\min(\gamma_1, 1)\lambda}\right)\right)}\right)\right)\|\mathbf{y}(i,j)\|_{\mathbf{V}^{-1}_{ser,t}}.$$

*We have $\mathcal{P}(\mathcal{I}) \geq 1 - \delta$.*

Moreover, when $i_{ser,\tau} = k^*$, we can trivially derive $\Delta(k^*, \hat{k}^*) = 0 \leq \epsilon$. The above discussion implies $\hat{k}^*$ output by FALinPE satisfies the $(\epsilon, \delta)$-condition (1).

We now continue to bound the sample complexity $\tau$. First, we need to establish Lemma 8 below, which upper bounds $T_{ser,\tau}(k)$, the number of observation on arm $k$ that is available to the server at $\tau$.

**Lemma 8.** *Under the setting of Theorem 2 and event $\mathcal{I}$, we can bound*

$$T_{ser,\tau}(k) \leq \max_{i,j \in \mathcal{A}} \frac{\rho(\mathbf{y}(i,j))p_k^*(\mathbf{y}(i,j))}{\max\left(\frac{\Delta(k^*,i)+\epsilon}{3}, \frac{\Delta(k^*,j)+\epsilon}{3}, \epsilon\right)^2} C_{ser,\tau}^2 + \gamma_2 M \sum_{s=1}^{K} T_{ser,\tau}(s), \, \forall k \in \mathcal{A}.$$

With Lemma 8, we can derive

$$\sum_{k=1}^{K} T_{ser,\tau}(k) \leq \sum_{k=1}^{K} \max_{i,j \in \mathcal{A}} \frac{\rho(\mathbf{y}(i,j))p_k^*(\mathbf{y}(i,j))}{\max\left(\frac{\Delta(k^*,i)+\epsilon}{3}, \frac{\Delta(k^*,j)+\epsilon}{3}, \epsilon\right)^2} C_{ser,\tau}^2 + \gamma_2 KM \sum_{s=1}^{K} T_{ser,\tau}(s)$$

$$\leq \frac{1}{1 - \gamma_2 KM} \sum_{k=1}^{K} \max_{i,j \in \mathcal{A}} \frac{\rho(\mathbf{y}(i,j))p_k^*(\mathbf{y}(i,j))}{\max\left(\frac{\Delta(k^*,i)+\epsilon}{3}, \frac{\Delta(k^*,j)+\epsilon}{3}, \epsilon\right)^2} C_{ser,\tau}^2.$$

Furthermore, based on the relationship between $\tau$ and $\sum_{k=1}^{K} T_{ser,\tau}(k)$ (Remark 4), we have

$$\begin{aligned}
\tau &= \sum_{k=1}^{K} T_{ser,\tau}(k) + \sum_{m=1}^{M} \sum_{k=1}^{K} T_{m,\tau}^{loc}(k) \\
&\leq \left(1 + \gamma_2 M\right) \sum_{k=1}^{K} T_{ser,\tau}(k) \\
&\leq \frac{1 + \gamma_2 M}{1 - \gamma_2 KM} \sum_{k=1}^{K} \max_{i,j \in \mathcal{A}} \frac{\rho(\mathbf{y}(i,j))p_k^*(\mathbf{y}(i,j))}{\max\left(\frac{\Delta(k^*,i)+\epsilon}{3}, \frac{\Delta(k^*,j)+\epsilon}{3}, \epsilon\right)^2} C_{ser,\tau}^2 \\
&= \frac{1 + \gamma_2 M}{1 - \gamma_2 KM} H_\epsilon^L C_{ser,\tau}^2
\end{aligned} \tag{41}$$

where the second inequality is owing to the inequality we establish above and the last equality is owing to the definition of $H_\epsilon^L$.

Recalling that we suppose $\gamma_1 = 1/(M^2)$, $\gamma_2 = 1/(2MK)$ and $0 < \lambda \leq \sigma^2\left(\sqrt{1 + \gamma_1 M} + \sqrt{2\gamma_1} M\right)^2 \log(2/\delta)$. We first need to decompose $C_{ser,\tau}$

$$\begin{aligned}
C_{ser,\tau} &= \sqrt{\lambda} + \left(\sqrt{2} + \sqrt{1 + \frac{1}{M}}\right)^2 \left(\sigma\sqrt{d \log\left(\frac{2}{\delta}\left(1 + \frac{(1+1/(2K))\sum_{k=1}^{K} T_{ser,\tau}(k)}{\lambda/M^2}\right)\right)}\right) \\
&= \sqrt{\lambda} + \left(\sqrt{2} + \sqrt{1 + \frac{1}{M}}\right)^2 \left(\sigma\sqrt{d \log\left(\frac{2}{\delta}\right) + d \log\left(1 + \frac{(1+1/(2K))\sum_{k=1}^{K} T_{ser,\tau}(k)}{\lambda/M^2}\right)}\right) \\
&\leq 2\left(\sqrt{2} + \sqrt{1 + \frac{1}{M}}\right)^2 \left(\sigma\sqrt{d \log\left(\frac{2}{\delta}\right) + d \log\left(1 + \frac{(1+1/(2K))\sum_{k=1}^{K} T_{ser,\tau}(k)}{\lambda/M^2}\right)}\right) \\
&\leq 2\left(\sqrt{2} + \sqrt{1 + \frac{1}{M}}\right)^2 \left(\sigma\sqrt{d \log\left(\frac{2}{\delta}\right) + d \log\left(1 + \frac{(1+1/(2K))\tau}{\lambda/M^2}\right)}\right).
\end{aligned} \tag{42}$$

The first inequality is owing to the definition of $\lambda$ and the last inequality is owing to $\sum_{k=1}^{K} T_{ser,\tau}(k) \leq \tau$. Substituting the last term of (42) into (41), we have

$$\tau \leq \frac{M + 1/(2K)}{M - 1/2} \left(\sqrt{2} + \sqrt{1 + \frac{1}{M}}\right)^2 H_\epsilon^L 4\sigma^2 d\left(\log\left(\frac{2}{\delta}\right) + \log\left(1 + \frac{(1+1/(2K))\tau}{\lambda/M^2}\right)\right).$$

We define

$$\Gamma = \frac{M + 1/(2K)}{M - 1/2} \left(\sqrt{2} + \sqrt{1 + \frac{1}{M}}\right)^2 H_\epsilon^L 4\sigma^2 d \log\left(\frac{2}{\delta}\right).$$

Let $\tau'$ be a parameter satisfies

$$\tau' \leq \tau = \frac{M+1/(2K)}{M-1/2}\left(\sqrt{2}+\sqrt{1+\frac{1}{M}}\right)^2 H_\epsilon^L 4\sigma^2 d\log\left(1+\frac{(1+1/(2K))\tau'}{\lambda/M^2}\right)+\Gamma$$

$$\leq \frac{M+1/(2K)}{M-1/2}\left(\sqrt{2}+\sqrt{1+\frac{1}{M}}\right)^2 H_\epsilon^L 4\sigma^2 d\sqrt{1+\frac{(1+1/(2K))\tau'}{\lambda/M^2}}+\Gamma.$$

We can further derive

$$\sqrt{\tau'} \leq \frac{M+1/(2K)}{M-1/2}\left(\sqrt{2}+\sqrt{1+\frac{1}{M}}\right)^2 H_\epsilon^L 4\sigma^2 d\sqrt{1+\frac{(1+1/(2K))\tau'}{\lambda/M^2}}+\Gamma = \Lambda. \tag{43}$$

In the light of (43), we can finally bound $\tau$ by

$$\tau \leq \frac{M+1/(2K)}{M-1/2}\left(\sqrt{2}+\sqrt{1+\frac{1}{M}}\right)^2 H_\epsilon^L 4\sigma^2 d\sqrt{1+\frac{(1+1/(2K))\Lambda^2}{\lambda/M^2}}+\Gamma. \tag{44}$$

**Lemma 9.** *Under the setting of Thoerem 2 and the communication strategy of line 11 in Algorithm 2, we can derive*

$$\mathbf{V}_{ser,t} \succeq (1/\gamma_1)\mathbf{V}_{m,t}^{loc}, \ \forall m \in \mathcal{M}, \ t \in [K+1,\tau]. \tag{45}$$

*Furthermore, following the results of Lemma 4, we can also derive*

$$\sum_{k=1}^{K} T_{ser,t}(k) \geq (1/\gamma_2)\sum_{k=1}^{K} T_{m,t}^{loc}(k), \ \forall m \in \mathcal{M}, \ t \in [K+1,\tau]. \tag{46}$$

**Proof for Lemma 7, Lemma 8, and Lemma 9** In the following paragraphs, we provide the detailed proof for the lemmas used above.

*Proof of Lemma 9.* The proof of (45) is similar to the proof in He et al. [2022]. Suppose the last round of agent $m$ communicates with the server is $t_1$ and the first event is triggered. Then, we can trivially derive

$$\mathbf{V}_{ser,t} \succ \mathbf{0} = \mathbf{V}_{m,t}^{loc}, \ \forall t \in [K+1,\tau].$$

Otherwise, according to the definition of the first event, we have

$$\det(\mathbf{V}_{m,t} + \mathbf{V}_{m,t}^{loc}) \leq (1+\gamma_1)\det(\mathbf{V}_{m,t}).$$

Based on the Lemma 17, we have

$$1+\gamma_1 \geq \frac{\det(\mathbf{V}_{m,t}+\mathbf{V}_{m,t}^{loc})}{\det(\mathbf{V}_{m,t})} \geq \frac{\|\mathbf{x}\|_{\mathbf{V}_{m,t}+\mathbf{V}_{m,t}^{loc}}^2}{\|\mathbf{x}\|_{\mathbf{V}_{m,t}}^2} = \frac{\|\mathbf{x}\|_{\mathbf{V}_{m,t}}^2 + \|\mathbf{x}\|_{\mathbf{V}_{m,t}^{loc}}^2}{\|\mathbf{x}\|_{\mathbf{V}_{m,t}}^2}$$

and

$$\mathbf{V}_{m,t} \succeq (1/\gamma_1)\mathbf{V}_{m,t}^{loc}.$$

With the fact that $\mathbf{V}_{ser,t} \succeq \mathbf{V}_{m,t}, \forall t \in [K+1,\tau]$, we can finish the proof of (45).

The proof of (46) is similar to the proof of Lemma 4. Combine the two results and we can finish the whole proof of Lemma 9. □

Based on Lemma 9, we can prove Lemma 7 as shown below.

*Proof of Lemma 7.* Following the same argument of Lemma 5, we only need to proof

$$\mathcal{P}(\mathcal{I}) = \mathcal{P}\left(\forall i,j \in \mathcal{A}, \forall t \in [K+1,\tau], \ |\hat{\Delta}_{ser,t}(i,j) - \Delta(i,j)| \leq \alpha_{ser,t}^L(i,j)\right) \geq 1 - \delta.$$

Decompose $|\hat{\Delta}_{ser,t}(i,j) - \Delta(i,j)|$, we have

$$\begin{aligned}
|\Delta(i,j) - \hat{\Delta}_{ser,t}(i,j)| &= |\mathbf{y}(i,j)^\top \boldsymbol{\theta}^* - \mathbf{y}(i,j)^\top \hat{\boldsymbol{\theta}}_{ser,t}| \\
&= |\mathbf{y}(i,j)^\top (\boldsymbol{\theta}^* - \hat{\boldsymbol{\theta}}_{ser,t})| \\
&\leq \|\mathbf{y}(i,j)\|_{\mathbf{V}_{ser,t}^{-1}} \|\boldsymbol{\theta}^* - \hat{\boldsymbol{\theta}}_{ser,t}\|_{\mathbf{V}_{ser,t}}.
\end{aligned}$$

Hence, according to the definition of the $\alpha_{ser,t}(i,j)$, we can derive

$$\mathcal{P}(\mathcal{I}) \geq \mathcal{P}\left(\forall t \in [K+1,\tau], \ \|\boldsymbol{\theta}^* - \hat{\boldsymbol{\theta}}_{ser,t}\|_{\mathbf{V}_{ser,t}} \leq C_{ser,t}\right) \tag{47}$$

and only need to proof

$$\mathcal{P}\left(\forall t \in [K+1,\tau], \ \|\boldsymbol{\theta}^* - \hat{\boldsymbol{\theta}}_{ser,t}\|_{\mathbf{V}_{ser,t}} \leq C_{ser,t}\right) \geq 1 - \delta. \tag{48}$$

We first discompose $\|\boldsymbol{\theta}^* - \hat{\boldsymbol{\theta}}_{ser,t}\|_{\mathbf{V}_{ser,t}}$

$$\begin{aligned}
\|\boldsymbol{\theta}^* - \hat{\boldsymbol{\theta}}_{ser,t}\|_{\mathbf{V}_{ser,t}} &= \|\boldsymbol{\theta}^* - \mathbf{V}_{ser,t}^{-1} \mathbf{b}_{ser,t}\|_{\mathbf{V}_{ser,t}} \\
&= \left\|\boldsymbol{\theta}^* - \mathbf{V}_{ser,t}^{-1}\left((\mathbf{V}_{ser,t} - \lambda\mathbf{I})\boldsymbol{\theta}^* + \sum_{m=1}^M \sum_{s=1}^{N_{m,t}} \mathbb{1}\{m_s = m\}\mathbf{x}_{m,s}\eta_{m,s}\right)\right\|_{\mathbf{V}_{ser,t}} \\
&= \left\|\boldsymbol{\theta}^* - \boldsymbol{\theta}^* + \lambda\mathbf{V}_{ser,t}^{-1}\boldsymbol{\theta}^* + \mathbf{V}_{ser,t}^{-1}\sum_{m=1}^M \sum_{s=1}^{N_{m,t}} \mathbb{1}\{m_s = m\}\mathbf{x}_{m,s}\eta_{m,s}\right\|_{\mathbf{V}_{ser,t}} \\
&= \|\lambda\mathbf{V}_{ser,t}^{-1}\boldsymbol{\theta}^*\|_{\mathbf{V}_{ser,t}} + \left\|\mathbf{V}_{ser,t}^{-1}\sum_{m=1}^M \sum_{s=1}^{N_{m,t}} \mathbb{1}\{m_s = m\}\mathbf{x}_{m,s}\eta_{m,s}\right\|_{\mathbf{V}_{ser,t}} \\
&= \lambda\|\boldsymbol{\theta}^*\|_{\mathbf{V}_{ser,t}^{-1}} + \left\|\mathbf{V}_{ser,t}^{-1}\left(\sum_{s=1}^t \mathbf{x}_{m_s,s}\eta_{m_s,s} - \sum_{m=1}^M \sum_{s=N_{m,t}+1}^t \mathbb{1}\{m_s = m\}\mathbf{x}_{m,s}\eta_{m,s}\right)\right\|_{\mathbf{V}_{ser,t}} \\
&\leq \sqrt{\lambda} + \underbrace{\left\|\mathbf{V}_{ser,t}^{-1}\sum_{s=1}^t \mathbf{x}_{m_s,s}\eta_{m_s,s}\right\|_{\mathbf{V}_{ser,t}}}_{\Lambda} + \underbrace{\left\|\mathbf{V}_{ser,t}^{-1}\sum_{m=1}^M \sum_{s=N_{m,t}+1}^t \mathbb{1}\{m_s = m\}\mathbf{x}_{m,s}\eta_{m,s}\right\|_{\mathbf{V}_{ser,t}}}_{\Gamma},
\end{aligned}$$
$$\tag{49}$$

where the last inequality is owing to $\mathbf{V}_{ser,t} \succeq \lambda\mathbf{I}$. We further decompose term $\Lambda$ and $\Gamma$. Based on the Lemma 18, we have $\forall t \in [K+1,\tau]$

$$\left\|\sum_{s=1}^t \mathbf{x}_{m_s,s}\eta_{m_s,s}\right\|_{\mathbf{V}_t^{all-1}} \leq \sigma\sqrt{d\log\left(\frac{2(1+t/\lambda)}{\delta}\right)} \tag{50}$$

holds with probability at least $1 - \delta/2$. With a union bound and utilize the self normalized martingale again, it holds that for each $m \in \mathcal{M}$ and $\forall t \in [K+1,\tau]$

$$\left\|\sum_{s=N_{m,t}+1}^t \mathbb{1}\{m_s = m\}\mathbf{x}_{m_s,s}\eta_{m_s,s}\right\|_{(\gamma_1\lambda\mathbf{I}+\mathbf{V}_{m,t}^{loc})^{-1}} \leq \sigma\sqrt{d\log\left(\frac{2(1+t/(\gamma_1\lambda))}{\delta}\right)} \tag{51}$$

holds with probability at least $1 - \delta/2$.

According to the Lemma 9 and (50), $\forall t \in [K+1, \tau]$, $\Lambda$ can be bounded by

$$
\left\| \mathbf{V}_{ser,t}^{-1} \sum_{s=1}^{t} \mathbf{x}_{m_s,s} \eta_{m_s,s} \right\|_{\mathbf{V}_{ser,t}}
$$

$$
= \left\| \sum_{s=1}^{t} \mathbf{x}_{m_s,s} \eta_{m_s,s} \right\|_{\mathbf{V}_{ser,t}^{-1}}
$$

$$
\leq \sqrt{1 + \gamma_1 M} \left\| \sum_{s=1}^{t} \mathbf{x}_{m_s,s} \eta_{m_s,s} \right\|_{\mathbf{V}_t^{all^{-1}}} \tag{52}
$$

$$
\leq \sqrt{1 + \gamma_1 M} \left( \sigma \sqrt{d \log \left( \frac{2(1 + t/\lambda)}{\delta} \right)} \right)
$$

with probability at least $1 - \delta/2$. The first inequality holds due to Lemma 9 and the last inequality holds according to (50). Besides, in the light of (51), $\forall t \in [K+1, \tau]$, $\Gamma$ can be bounded by

$$
\left\| \mathbf{V}_{ser,t}^{-1} \sum_{m=1}^{M} \sum_{s=N_{m,t}+1}^{t} \mathbb{1}\{m_s = m\} \mathbf{x}_{m,s} \eta_{m,s} \right\|_{\mathbf{V}_{ser,t}}
$$

$$
= \left\| \sum_{m=1}^{M} \sum_{s=N_{m,t}+1}^{t} \mathbb{1}\{m_s = m\} \mathbf{x}_{m,s} \eta_{m,s} \right\|_{\mathbf{V}_{ser,t}^{-1}}
$$

$$
\leq \sum_{m=1}^{M} \left\| \sum_{s=N_{m,t}+1}^{t} \mathbb{1}\{m_s = m\} \mathbf{x}_{m,s} \eta_{m,s} \right\|_{\mathbf{V}_{ser,t}^{-1}} \tag{53}
$$

$$
\leq \sqrt{2\gamma_1} \sum_{m=1}^{M} \left\| \sum_{s=N_{m,t}+1}^{t} \mathbb{1}\{m_s = m\} \mathbf{x}_{m,s} \eta_{m,s} \right\|_{(\mathbf{V}_{m,t}^{loc} + \gamma_1 \lambda \mathbf{I})^{-1}}
$$

$$
\leq \sqrt{2\gamma_1} M \left( \sigma \sqrt{d \log \left( \frac{2(1 + t/(\gamma_1 \lambda))}{\delta} \right)} \right)
$$

with probability at least $1 - \delta/2$. The second inequality holds is due to $\forall m \in \mathcal{M}$

$$
\mathbf{V}_{ser,t} \succeq \frac{1}{\gamma_1} \mathbf{V}_{m,t}^{loc}
$$

$$
\frac{1}{2} \mathbf{V}_{ser,t} + \frac{1}{2} \mathbf{V}_{ser,t} \succeq \frac{1}{2} \lambda \mathbf{I} + \frac{1}{2\gamma_1} \mathbf{V}_{m,t}^{loc}.
$$

Combine (49), (52) and (53), due to the server or the agents can not directly derive $t$, we can utilize $(1 + \gamma_2 M) \sum_{k=1}^{K} T_{ser,t}(k)$ to replace $t$ and the above inequalities still hold. We can finally get $\forall t \in [K+1, \tau]$

$$
\|\boldsymbol{\theta}^* - \hat{\boldsymbol{\theta}}_{ser,t}\|_{\mathbf{V}_{ser,t}} \leq \sqrt{\lambda} + \left( \sqrt{2\gamma_1} M + \sqrt{1 + \gamma_1 M} \right) \left( \sigma \sqrt{d \log \left( \frac{2}{\delta} \left( 1 + \frac{(1 + \gamma_2 M) \sum_{k=1}^{K} T_{ser,t}(k)}{\min(\gamma_1, 1) \lambda} \right) \right)} \right)
$$

holds with probability at least $1 - \delta$. Combine this with (48), here we finish the proof of Lemma 7. $\qquad \square$

Before proving Lemma 8, we first need to establish Lemma 10 and Lemma 11 below.

**Lemma 10.** *Following the setting of Theorem 2, $\forall t \in [K+1, \tau]$, The matrix norm $\|\mathbf{y}(i,j)\|_{\mathbf{V}_{m_t,t}^{-1}}$ can be bounded by*

$$
\|\mathbf{y}(i,j)\|_{V_{m_t,t}^{-1}} \leq \sqrt{\frac{\rho(\mathbf{y}(i,j))}{T_{m_t,t}(i,j)}}, \text{ and } \|\mathbf{y}(i,j)\|_{V_{ser,t}^{-1}} \leq \sqrt{\frac{\rho(\mathbf{y}(i,j))}{T_{ser,t}(i,j)}}, \forall i, j \in \mathcal{A}, \tag{54}
$$

*where*

$$
T_{m_t,t}(i,j) = \min_{k \in \mathcal{A}, \, p_k^*(\mathbf{y}(i,j)) > 0} T_{m_t,t}(k)/p_k^*(\mathbf{y}(i,j))
$$

$$
T_{ser,t}(i,j) = \min_{k \in \mathcal{A}, \, p_k^*(\mathbf{y}(i,j)) > 0} T_{ser,t}(k)/p_k^*(\mathbf{y}(i,j)). \tag{55}
$$

*Proof of Lemma 10.* According to the Lemma 2 of Xu et al. [2017], the optimal value of (13) (i.e., $\rho(\mathbf{y}(i,j))$) is equal to the optimal value of

$$\min_{p_k, w_k} \sum_{k=1}^{K} \frac{w_k^2}{p_k}$$

$$s.t. \quad \mathbf{y}(i,j) = \sum_{k=1}^{K} w_k \mathbf{x}_k \tag{56}$$

$$\sum_{k=1}^{K} p_k = 1, \ p_k > 0, \ w_k \in \mathcal{R},$$

for all $i, j \in \mathcal{A}$.

Due to $\mathbf{V}_{m_t, t}$ and $T_{m_t, t}(k)$, $\forall k \in \mathcal{A}$, $t \in [K+1, \tau]$ are all downloaded from the server. This implies $\forall t_1 \in [K+1, \tau]$, there exists a $t_2 \in [K+1, \tau]$ which satisfies

$$\mathbf{V}_{m_{t_1}, t_1} = \mathbf{V}_{ser, t_2} \text{ and } T_{m_{t_1}, t_1}(k) = T_{ser, t_2}(k), \ \forall k \in \mathcal{A}.$$

Therefore, we only need to prove the second inequality of (54). We can decompose the covariance matrix $\mathbf{V}_{ser, t} = \lambda \mathbf{I} + \sum_{k=1}^{K} T_{ser, t}(k) \mathbf{x}_k \mathbf{x}_k^\top$. We define the auxiliary covariance matrix as $\tilde{\mathbf{V}}_{ser, t} = \lambda \mathbf{I} + \sum_{k=1}^{K} T_{ser, t}(i, j) p_k^*(\mathbf{y}(i, j)) \mathbf{x}_k \mathbf{x}_k^\top$. From (55), we have

$$T_{ser, t}(i, j) p_k^*(\mathbf{y}(i, j)) \leq T_{ser, t}(k), \ \forall k, i, j \in \mathcal{A}$$

which implies $\tilde{\mathbf{V}}_{ser, t} \preceq \mathbf{V}_{ser, t}$ and

$$\mathbf{y}(i, j)^\top \mathbf{V}_{ser, t}^{-1} \mathbf{y}(i, j) \leq \mathbf{y}(i, j)^\top \tilde{\mathbf{V}}_{ser, t}^{-1} \mathbf{y}(i, j), \ i, j \in \mathcal{A}.$$

We then bound $\mathbf{y}(i, j)^\top \tilde{\mathbf{V}}_{ser, t}^{-1} \mathbf{y}(i, j)$, according to the KKT condition of (56), we have the following formulas

$$w_k^*(\mathbf{y}(i, j)) = \frac{1}{2} p_k^*(\mathbf{y}(i, j)) \mathbf{x}_k^\top \varepsilon, \ \forall k, i, j \in \mathcal{A}$$

$$\mathbf{y}(i, j) = \frac{1}{2} \sum_{k=1}^{K} p_k^*(\mathbf{y}(i, j)) \mathbf{x}_k \mathbf{x}_k^\top \varepsilon, \ \forall i, j \in \mathcal{A},$$

where $\varepsilon \in \mathcal{R}^d$ corresponds to the Lagrange multiplier. Hence, we can rewrite $\mathbf{y}(i, j)^\top \tilde{\mathbf{V}}_{ser, t}^{-1} \mathbf{y}(i, j)$ as

$$\mathbf{y}(i, j)^\top \tilde{\mathbf{V}}_{ser, t}^{-1} \mathbf{y}(i, j) = \frac{1}{4} \left( \sum_{k=1}^{K} p_k^*(\mathbf{y}(i, j)) \mathbf{x}_k \mathbf{x}_k^\top \varepsilon \right)^\top \tilde{\mathbf{V}}_{ser, t}^{-1} \left( \sum_{k=1}^{K} p_k^*(\mathbf{y}(i, j)) \mathbf{x}_k \mathbf{x}_k^\top \varepsilon \right). \tag{57}$$

Besides, based on (56), we can rewrite $\rho(\mathbf{y}(i, j))$ as

$$\rho(\mathbf{y}(i, j)) = \sum_{k=1}^{K} \frac{w_k^{*2}(\mathbf{y}(i, j))}{p_k^*(\mathbf{y}(i, j))} = \frac{1}{4} \varepsilon^\top \left( \sum_{k=1}^{K} p_k^*(\mathbf{y}(i, j)) \mathbf{x}_k \mathbf{x}_k^\top \right) \varepsilon. \tag{58}$$

In the light of (57) and (58), we can bound $\mathbf{y}(i, j)^\top \tilde{\mathbf{V}}_{ser, t}^{-1} \mathbf{y}(i, j) - \rho(\mathbf{y}(i, j)) / T_{ser, t}(i, j)$ with 0

$$\mathbf{y}(i, j)^\top \tilde{\mathbf{V}}_{ser, t}^{-1} \mathbf{y}(i, j) - \frac{\rho(\mathbf{y}(i, j))}{T_{ser, t}(i, j)}$$

$$= \frac{1}{4} \varepsilon^\top \left( \left( \sum_{k=1}^{K} p_k^*(\mathbf{y}(i, j)) \mathbf{x}_k \mathbf{x}_k^\top \right) - \frac{\tilde{\mathbf{V}}_{ser, t}}{T_{ser, t}(i, j)} \right) \tilde{\mathbf{V}}_{ser, t}^{-1} \left( \sum_{k=1}^{K} p_k^*(\mathbf{y}(i, j)) \mathbf{x}_k \mathbf{x}_k^\top \right) \varepsilon$$

$$= -\frac{\lambda}{4} \varepsilon^\top \tilde{\mathbf{V}}_{ser, t}^{-1} \left( \sum_{k=1}^{K} p_k^*(\mathbf{y}(i, j)) \mathbf{x}_k \mathbf{x}_k^\top \right) \varepsilon$$

$$\leq 0.$$

The second equality holds due to the definition of the $\tilde{\mathbf{V}}_{ser, t}$, and the last inequality holds due to $\lambda > 0$ and the definition of the positive definite matrix. Here we finish the proof of Lemma 10. $\square$

**Lemma 11.** *Under the setting of Theorem 2 and event $\mathcal{I}$, $\forall t \in [K+1, \tau]$, $B(t)$ can be bounded as follows*

$$B(t) \leq \min\left(0, -\max\left(\Delta(k^*, i_{ser,t}), \Delta(k^*, j_{ser,t})\right) + 2\alpha_{ser,t}^L(i_{ser,t}, j_{ser,t})\right) + \alpha_{ser,t}^L(i_{ser,t}, j_{ser,t}).$$

*Proof of Lemma 11.* This proof is similar to the proof of Lemma 5. According to the definition of the event $\mathcal{I}$, consider the case when $i_{ser,t} = k^*$, we have

$$
\begin{aligned}
B(t) &= \hat{\Delta}_{ser,t}(j_{ser,t}, i_{ser,t}) + \alpha_{ser,t}^L(i_{ser,t}, j_{ser,t}) \\
&\leq \Delta(j_{ser,t}, i_{ser,t}) + 2\alpha_{ser,t}^L(i_{ser,t}, j_{ser,t}) \\
&= -\Delta(k^*, j_{ser,t}) + 2\alpha_{ser,t}^L(i_{ser,t}, j_{ser,t}).
\end{aligned}
\tag{59}
$$

Consider the case when $j_{ser,t} = k^*$, we have

$$
\begin{aligned}
B(t) &= \hat{\Delta}_{ser,t}(j_{ser,t}, i_{ser,t}) + \alpha_{ser,t}^L(i_{ser,t}, j_{ser,t}) \\
&\leq -\hat{\Delta}_{ser,t}(j_{ser,t}, i_{ser,t}) + \alpha_{ser,t}^L(i_{ser,t}^L, j_{ser,t}) \\
&\leq -\Delta(j_{ser,t}, i_{ser,t}) + 2\alpha_{ser,t}^L(i_{ser,t}, j_{ser,t}) \\
&= -\Delta(k^*, i_{ser,t}) + 2\alpha_{ser,t}^L(i_{ser,t}, j_{ser,t}),
\end{aligned}
\tag{60}
$$

where the first inequality is owing to $\hat{\Delta}_{ser,t}(j_{ser,t}, i_{ser,t}) \leq 0$.

Combine (59) and (60), it yields

$$B(t) \leq \min\left(0, -\max\left(\Delta(k^*, i_{ser,t}), \Delta(k^*, j_{ser,t})\right) + \alpha_{ser,t}^L(i_{ser,t}, j_{ser,t})\right) + \alpha_{ser,t}^L(i_{ser,t}, j_{ser,t}) \tag{61}$$

when $i_{ser,t} = k^*$ or $j_{ser,t} = k^*$.

Consider the case when $i_{ser,t} \neq k^*$ and $j_{ser,t} \neq k^*$, then we can derive

$$
\begin{aligned}
B(t) &= \hat{\Delta}_{ser,t}(j_{ser,t}, i_{ser,t}) + \alpha_{ser,t}^L(i_{ser,t}, j_{ser,t}) \\
&\leq \Delta(j_{ser,t}, k^*) + \Delta(k^*, i_{ser,t}) + 2\alpha_{ser,t}^L(i_{ser,t}, j_{ser,t}) \\
&\leq \Delta(j_{ser,t}, k^*) + 3\alpha_{ser,t}^L(i_{ser,t}, j_{ser,t}) \\
&= -\Delta(k^*, j_{ser,t}) + 3\alpha_{ser,t}^L(i_{ser,t}, j_{ser,t})
\end{aligned}
\tag{62}
$$

where the second inequality holds is owing to

$$
\begin{aligned}
\alpha_{ser,t}^L(i_{ser,t}, j_{ser,t}) &\geq \hat{\Delta}_{ser,t}(j_{ser,t}, i_{ser,t}) + \alpha_{ser,t}^L(i_{ser,t}, j_{ser,t}) \\
&\geq \hat{\Delta}_{ser,t}(k^*, i_{ser,t}) + \alpha_{ser,t}^L(i_{ser,t}, k^*) \\
&\geq \Delta(k^*, i_{ser,t}).
\end{aligned}
$$

We also can show

$$
\begin{aligned}
B(t) &= \hat{\Delta}_{ser,t}(j_{ser,t}, i_{ser,t}) + \alpha_{ser,t}^L(i_{ser,t}, j_{ser,t}) \\
&\leq \alpha_{ser,t}^L(i_{ser,t}, j_{ser,t}) \\
&\leq -\Delta(k^*, i_{ser,t}) + \hat{\Delta}_{ser,t}(k^*, i_{ser,t}) + \alpha_{ser,t}^L(k^*, i_{ser,t}) + \alpha_{ser,t}^L(i_{ser,t}, j_{ser,t}) \\
&\leq -\Delta(k^*, i_{ser,t}) + \hat{\Delta}_{ser,t}(j_{ser,t}, i_{ser,t}) + \alpha_{ser,t}^L(i_{ser,t}, j_{ser,t}) + \alpha_{ser,t}^L(i_{ser,t}, j_{ser,t}) \\
&\leq -\Delta(k^*, i_{ser,t}) + 2\alpha_{ser,t}^L(i_{ser,t}, j_{ser,t}).
\end{aligned}
\tag{63}
$$

The second inequality is due to the definition of the even $\mathcal{I}$ and the third inequality is due to the definition of $j_{ser,t}$.

Combine (62) and (63), it yields

$$B(t) \leq \min\left(0, -\Delta(k^*, k_{ser,t}) + 2\alpha_{ser,t}^L(i_{ser,t}, j_{ser,t})\right) + \alpha_{ser,t}^L(i_{ser,t}, j_{ser,t}). \tag{64}$$

Combine the (61) and (64), then we can finish the proof of Lemma 11. □

*Proof of Lemma 8.* The difference between this proof and Lemma 3 is Algorithm 2 employs a different arm selection strategy. We define at round $t_k \neq \tau$, an agent $m$ communicates with the server and $k_{ser,t_k} = k$, where

$$k_{ser,t_k} = \arg\min_{k \in \mathcal{A}} \frac{T_{ser,t_k}(k)}{p_k^*(\mathbf{y}(i_{ser,t}, j_{ser,t}))}. \tag{65}$$

And from round $t \in [t_k + 1, \tau]$, when any agent communicating with the server, $k_{ser,t} \neq k$ . This implies

$$T_{ser,\tau}(k) \leq T_{m,t_k+1}(k) + (\gamma_2 M) \sum_{s=1}^{K} T_{ser,\tau}(s)$$

$$= T_{ser,t_k}(k) + (\gamma_2 M) \sum_{s=1}^{K} T_{ser,\tau}(s).$$

The inequality holds due to for $t \in [t_k + 1, \tau]$, $\forall m \in \mathcal{M}$ would upload $T_{m,t}^{loc}(k) > 0$ to the server at most one time and $T_{m,t}^{loc}(k) \leq \gamma_2 \sum_{s=1}^{K} T_{ser,\tau}(s)$ according to the Lemma 9.

With Lemma 11, we can derive

$$\epsilon \leq B(t_k)$$
$$\leq \min\left(0, -\max(\Delta(k^*, i_{ser,t_k}), \Delta(k^*, j_{ser,t_k})) + 2\alpha_{ser,t_k}^L(i_{ser,t_k}, j_{ser,t_k})\right) + \alpha_{ser,t_k}^L(i_{ser,t_k}, j_{ser,t_k}). \tag{66}$$

We would further bound $T_{ser,t}(k)$. Recalling the arm selection strategy of Algorithm 2, when $k$ is chosen by agent $m$ in round $t_k + 1$, this implies

$$T_{ser,t_k}(i_{ser,t_k}, j_{ser,t_k}) = T_{ser,t_k}(k)/p_k^*(i_{ser,t_k}, j_{ser,t_k}). \tag{67}$$

Recalling the definition of $\alpha_{ser,t_k}(i_{ser,t_k}, j_{ser,t_k})$ and substituting (54) and (66) into (67), we can derive

$$T_{ser,t_k}(k) \leq \frac{\rho(\mathbf{y}(i_{ser,t_k}, j_{ser,t_k}))p_k^*(\mathbf{y}(i_{ser,t_k}, j_{ser,t_k}))}{\max\left(\frac{\Delta(k^*, i_{ser,t_k})+\epsilon}{3}, \frac{\Delta(k^*, j_{ser,t_k})+\epsilon}{3}, \epsilon\right)^2} C_{ser,t_k}^2$$

$$\leq \max_{i,j \in \mathcal{A}} \frac{\rho(\mathbf{y}(i,j))p_k^*(\mathbf{y}(i,j))}{\max\left(\frac{\Delta(k^*, i)+\epsilon}{3}, \frac{\Delta(k^*, j)+\epsilon}{3}, \epsilon\right)^2} C_{ser,\tau}^2.$$

We can finally bound $T_{ser,\tau}(k)$, i.e.

$$T_{ser,\tau}(k) \leq \max_{i,j \in \mathcal{A}} \frac{\rho(\mathbf{y}(i,j))p_k^*(\mathbf{y}(i,j))}{\max\left(\frac{\Delta(k^*, i)+\epsilon}{3}, \frac{\Delta(k^*, j)+\epsilon}{3}, \epsilon\right)^2} C_{ser,\tau}^2 + \gamma_2 M \sum_{s=1}^{K} T_{ser,\tau}(s).$$

Here we finish the proof of Lemma 8. $\qquad\square$

## D   AUXILIARY LEMMAS

**Lemma 12** (Conditionally $\sigma$-sub-Gaussian noise [Abbasi-Yadkori et al., 2011, Lattimore and Szepesvári, 2020, Li and Wang, 2022a, He et al., 2022]). *The noise $\eta_{m_t,t}$ of the linear case is drawn from a conditionally $\sigma$-sub-Gaussian distribution, which satisfies*

$$\mathbb{E}\left[e^{\lambda \eta_{m_t,t}} \Big| \mathbf{x}_{m_1,1}, ..., \mathbf{x}_{m_t,t}, m_1, ..., m_t, \eta_{m_1,1}, ..., \eta_{m_t,t}\right] \leq e^{\sigma^2 \lambda^2/2}, \quad \forall \lambda \in \mathcal{R}. \tag{68}$$

**Lemma 13** (Hoffeding inequality). *Suppose $X_1, X_2, ..., X_n$ are i.i.d drawn from a $\sigma$-sub-Gaussian distribution and $\bar{X} = (1/n)\sum_{s=1}^{n} X_s$ represents the mean, then*

$$\mathcal{P}(|\mathbb{E}[X] - \bar{X}| \geq -a) \leq e^{-a^2 n/2\sigma^2}.$$

**Lemma 14** (Lemma 10 of Abbasi-Yadkori et al. [2011]). *The matrix norm can be bounded by*

$$det\Big(\lambda\mathbf{I} + \sum_{s=1}^{t}\mathbf{x}_{m_s,s}\mathbf{x}_{m_s,s}^{\top}\Big) \leq \Big(\lambda + \frac{t}{d}\Big)^{d}.$$

**Lemma 15** (Lemma 2.3 of Tie et al. [2011]). *For arbitrary positive definitive matrices A, B and C, it has*

$$det(A + B + C)det(A) \leq det(A + B)det(A + C).$$

**Lemma 16** (Lemma 2.2 of Tie et al. [2011]). *For arbitrary positive definitive matrices A, B and C, it has*

$$det(A + B + C) + det(A) \geq det(A + B) + det(A + C).$$

**Lemma 17** (Lemma 12 of Abbasi-Yadkori et al. [2011]). *For arbitrary positive definitive matrices A and B satisfies $A \succ B$, it has*

$$\frac{\|\mathbf{x}\|_A^2}{\|\mathbf{x}\|_B^2} \leq \frac{det(A)}{det(B)}.$$

**Lemma 18** (Theorem 1 of Abbasi-Yadkori et al. [2011]). *For $t \in |t|$, it has*

$$\Big\|\sum_{s=1}^{t}\mathbf{x}_{m_s,s}\eta_{m_s,s}\Big\|_{\lambda\mathbf{I}+\sum_{s=1}^{t}\mathbf{x}_{m_s,s}\mathbf{x}_{m_s,s}^{\top}} \leq \sigma\sqrt{d\log\Big(\frac{1+t/\lambda}{\delta}\Big)}$$

*holds with probability at least $1 - \delta$.*

