# OpenReview forum: "Pure Exploration in Asynchronous Federated Bandits"
_auai.org/UAI/2024/Conference — UAI 2024 poster_

### Official Review · Reviewer_Mvsf · 2024-02-28

**Q2-1 Originality-Novelty:** 2
**Q2-2 Correctness-Technical Quality:** 2
**Q2-5 Clarity Of Writing:** 3

**Q1 Summary And Contributions:**

This draft tries to study Pure Exploration in Asynchronous Federated in stochastic and linear Bandits settings, where two algorithms are proposed

**Q2-3 Extent To Which Claims Are Supported By Evidence:**

2: Fair: the main claims are somewhat supported by evidence (but the experimental evaluation may be weak, or does not match entirely with the claims, important baselines may be missing, proofs contain important ideas but lack rigor, algorithmic details are only discussed superficially, references are imprecise, assumptions are not sufficiently motivated or explicated, etc.).

**Q2-4 Reproducibility:**

3: Good: key resources (e.g. proofs, code, data) are available and key details (e.g. proofs, experimental setup) are sufficiently well-described for competent researchers to confidently reproduce the main results.

**Q3 Main Strengths:**

1. this draft is easy to understand and follow
2. combine federated learning with MABs is important

**Q4 Main Weakness:**

1. your empirical performance is not impressive, and how this proposal is useful in practice or industry is unclear, which should be clearly articulated
2. in your experiments, the data scale is too small and you don't have large-scale and real-world or production data based experimental results to support your claims which is one of the main drawbacks of this work

**Q5 Detailed Comments To The Authors:**

This manuscript studies asynchronous federated pure exploration algorithms, with the potential to incorporate decentralised bandits, there are related state-of-the-art you should compare: Fast Distributed Bandits for Online Recommendation Systems, Distributed Online and Bandit Convex Optimization

**Q9 Complying With Reviewing Instructions:**

Yes

---

> ### Author Rebuttal · Authors · 2024-04-05
>
> We thank the reviewer for the positive comments on the presentation and significance of our paper, and the constructive questions to help further clarify our designs and algorithms.
>
> **Q1:** How this proposal is useful in practice or industry is unclear, which should be clearly articulated.
>
> **A1:** Thank you for the suggestion. We here propose a practical example of our algorithms.
>
> Let's consider a sequential experimental design problem, e.g., for drug discovery or chemical synthesis, where our goal is to identify an arm that is $\epsilon$-near optimal (i.e., chemical with desired properties) with high probability. In this problem, we are not concerned about cumulative regret (i.e., the quality of the chemicals tried during the online learning process); instead, we only care about whether we can find the optimal arm **in the end**, and the corresponding sample complexity and communication cost due to their expensive nature (see the introduction in [Hillel et al., 2013; R'eda, 2022; Du et al., 2021] for details). Additionally, each laboratory lacks samples (i.e., funding for resource) to complete the task individually, so we need to involve multiple labs to collaborate on the learning task. These requirements motivate people to study federated pure exploration problems. Besides, previous synchronous federated pure exploration algorithms assume every agent (i.e., lab) should participate in the exploration (i.e., do the experiment) in each round and the server can force all the agents to upload their data in synchronization rounds. This is impractical due to some agents may get offline (e.g., they run out of resources), and all other agents should wait until they get online (e.g., collect enough resources), this will significantly reduce the learning speed (see the introduction in [He et al., 2022; Li et al., 2021] for details). Our asynchronous federated pure exploration algorithms can alleviate these assumptions: 1) Each agent can decide whether to participate in each round. Full participation isn't obligatory, thus accommodating temporarily offline agents; and 2) communication between each agent and the server is asynchronous and completely independent of other agents. Based on our discussion, we believe our asynchronous federated pure exploration algorithms are more practical than the previous synchronous federated pure exploration algorithms.
>
> We will add this example in the future version of our paper.
>
> **Q2:** In your experiments, the data scale is too small and you don't have large-scale and real-world or production data based experimental results to support your claims.
>
> **A2:** Thank you for your comment and suggestion. The choice of arm numbers (i.e., $K = 5$ and $10$) in our experimental section aligns with common practice in other papers [Mitra et. al., 2021; Du et al., 2021]. A production system like recommender system usually contains two or more stages where first stage will filter out most of the low reward arms and leaves tens or hundreds of candidates, so that the later stages focus solely on identifying the best arm among those promising ones (with small reward gap). Additionally, we note that we present experimental results using real-world MovieLens dataset for the linear case in  Appendix A. We intend to integrate this result into the main paper in final version with additional pages allowed.
>
> **Q3:** This manuscript has the potential to incorporate decentralised bandits, there are related state-of-the-art you should compare: Fast Distributed Bandits for Online Recommendation Systems, Distributed Online and Bandit Convex Optimization.
>
> **A3:** Thank you suggesting the relevant papers. We will add them to our reference.
>
> The paper "Fast Distributed Bandits for Online Recommendation Systems" introduces a novel distributed bandit-based algorithm **DistCLUB**. This algorithm lazily forms clusters in a distributed manner, substantially reducing the need for network data sharing and achieving high scalability. Besides, the paper "Distributed Online and Bandit Convex Optimization" aims to minimize regret on $M$ machines working in parallel over $T$ rounds with $R$ intermittent communication budget and bandit feedback. However, these algorithms can only work in the synchronous environment, in our paper, we consider the more general asynchronous environment and propose two algorithms that can achieve near-optimal theoretical performance in such environment.
>
> This discussion will also be included in the paper.
>
> **Reference:**
>
> [1] Du. (2021). Collaborative Pure Exploration in Kernel Bandit.
> [2] Mitra. (2021). Exploiting Heterogeneity in Robust Federated Best-Arm Identification.
> [3] He. (2022). A Simple and Provably Efficient Algorithm for Asynchronous Federated Linear Bandits.
> [4] Li. (2021). Asynchronous UCB Algorithms for Federated Linear Bandits.
> [5] Hillel. (2013). Distributed Exploration in Multi-Armed Bandits.
> [6] R'eda. (2022). Near-Optimal Collaborative Learning in Bandits.

---

### Official Review · Reviewer_DTya · 2024-03-14

**Q2-1 Originality-Novelty:** 3
**Q2-2 Correctness-Technical Quality:** 3
**Q2-5 Clarity Of Writing:** 3

**Q1 Summary And Contributions:**

This paper studies a pure exploration problem under the federated setting, where the goal is to output a near-optimal arm with sample complexity and communication cost as less as possible. Further, this paper assumes that all the agents are asynchronous, i.e., in each time step, only one agent is active, and there is no contraint on how these agents become active (e.g., an agent can only be active once and never appears again). In this case, the authors propose algorithms for both the MAB model and the linear bandit model. Theoratical analysis shows that the both the algorithms achieves a near optimal sample complexity, and the communication costs are also limited. Empirical results also demonstrate the effectiveness of the algorithms.

**Q2-3 Extent To Which Claims Are Supported By Evidence:**

4: Excellent: all claims are supported by very convincing evidence (in the form of comprehensive experimental evaluation, rigorous mathematical proofs, detailed (pseudo-)code, precise references, well-motivated and realistic assumptions) and the authors deliver what they promise.

**Q2-4 Reproducibility:**

4: Excellent: key resources (e.g. proofs, code, data) are available and key details (e.g. proof sketches, experimental setup) are comprehensively described for competent researchers to confidently and easily reproduce the main results.

**Q3 Main Strengths:**

- The problem setting is well motivated.
- The complexity bounds are near optimal.
- The communication costs are not very large.
- There are empirical results demonstrating the effectiveness of the algorithms.

**Q4 Main Weakness:**

- Some parts of the paper are not clear enough (please see below for details).

**Q5 Detailed Comments To The Authors:**

- Why the communication costs (in both cases) do not dependent on the confidence level $\delta$? For example, in eq (18), it is shown that $C(\tau) \le 2(M+1/\gamma)\log \tau$. However, since $\tau$ depends on $\delta$ (with order $\log{1\over \delta}$), the communication cost should also depend on $\delta$ (with order $\log \log{1\over \delta}$).

- In FALinPE, why we require a hybrid event-triggered strategy, specifically, why we want to upload/download if the local observation number is higher than $\gamma_2$ times the server observation number? I guess it is used to control the $1 + M\gamma_2$ term in the confidence radius. However, since this term is in log, could we do this kind of upload/download in a much less frequency? Maybe this can decrease the communication cost?

- In your experiments (a), why the communication cost of FAMABPE almost remains the same under different sample complexity $\tau$?

**Q9 Complying With Reviewing Instructions:**

Yes

---

> ### Author Rebuttal · Authors · 2024-04-05
>
> We thank the reviewer for the positive comments on the quality of our work, and the constructive questions to help further clarify our designs and algorithms.
>
> **Q1:** Why the communication costs (in both cases) do not dependent on the confidence level $\delta$?
>
>  **A1:** Thank you for your insightful question. In Theorem $1$, we bound the sample complexity of **FAMABPE** as $\tau = O(H_\epsilon^M \log(H_\epsilon^M/\delta) )$ and the sample complexity as $C(\tau) = 2(M + 1/\gamma)\log(\tau)$. By substituting the first bound into the second, we derive $C(\tau) = O((M + 1/\gamma)\log(H_\epsilon^M \log(H_\epsilon^M/\delta)))$, which is related to $\delta$ with an order of $\log(\log(1/\delta))$. The same reasoning applies to Theorem $2$. Our current version omits the $\log(\log(.))$ term for the sake of simplicity in expression. We will add the $\delta$-dependent results to the future version of our paper.
>
>  **Q2:** In **FALinPE**, why we require a hybrid event-triggered strategy, specifically, why we want to upload/download if the local observation number is higher than $\gamma_2$ times the server observation number? I guess it is used to control the term $(1 + M\gamma_2)$ in the confidence radius. However, since this term is in $\log$, could we do this kind of upload/download in a much less frequency? Maybe this can decrease the communication cost?
>
> **A2:** Thank you for your helpful question. There are two reasons that we need to design the communication protocol with event $2$.
>
> The first one is neither the agents nor the server have access to the global observation number (i.e., time index $t$).  Consequently, we cannot directly employ the time index $t$ to establish exploration bonuses in the asynchronous environment. Previous works by [Li et. al., 2021, 2023; He et. al., 2022] encountered similar constraints, but they suppose the time horizon $T$ of the regret minimization problem is known and can utilize $T$ to establish the exploration bonus. However, the time horizon $\tau$ in the fixed confidence pure exploration problem is unavailable. Hence, we design an upper bound for $t$ by leveraging the triggered event (i.e., event $2$ in the hybrid triggered strategy) to devise the exploration bonus. In Lemma $4$ and the proof of Lemma $2$, we demonstrate that $t  \le (1+\gamma M)\sum_{k=1}^K T_{ser,t}(k)$. Subsequently, we substitute $(1+\gamma M)\sum_{k=1}^K T_{ser,t}(k)$ and $(1+\gamma M)\sum_{k=1}^K T_{m_t,t}(k)$ into the exploration bonuses of the server and $m_t$ to replace $t$.
>
> The second reason is that when the server terminates the algorithm, some agents may possess data that has not been uploaded to the server. We wish the amount of these data to be small compared with the sample complexity $\tau$ since they have no contribution to identifying the estimated best arm. The event $2$ can efficiently limit the number of these useless samples.
>
> The above detailed discussion will be added to the future version of our paper.
>
> **Q3:** In your experiments (a), why the communication cost of **FAMABPE** almost remains the same under different sample complexity $\tau$?
>
> **A3:** Thank you for your question. We think the reason is that the communication cost is only logarithmically related to the sample complexity $\tau$ (i.e., $C(\tau) = O((1 + 1/\gamma)\log(\tau))$). Here is the data (the average communication cost of $10$ runs) of the communication cost of **FAMABPE** in Fig 1 (a):
>
> |gap value:|gap = 0.1| gap = 0.2 | gap = 0.3 | gap = 0.4 | gap = 0.5 |
> |------|----|----|----|----|----|
> |    average cost:     |   210.2   |   176.6   |   157.6   |   145   |   133.8   |
>
> From the data above, it is evident that as the sample complexity decreases gradually, the corresponding communication cost also decreases gradually. We apologize for the unclear trend in the current Fig 1 (a) due to the scale of the figure. We will annotate each data point in Fig 1 (a) in the new version of the paper.
>
> **Reference:**
>
> [1] He, J., Wang, T., Min, Y., $\And$ Gu, Q. (2022). A Simple and Provably Efficient Algorithm for Asynchronous Federated Contextual Linear Bandits. ArXiv, abs/2207.03106.
> [2] Li, C., $\And$ Wang, H. (2021). Asynchronous Upper Confidence Bound Algorithms for Federated Linear Bandits. ArXiv, abs/2110.01463.
> [3] Li, C., Wang, H., Wang, M., $\And$ Wang, H. (2023). Learning Kernelized Contextual Bandits in a Distributed and Asynchronous Environment. International Conference on Learning Representations.

---

### Official Review · Reviewer_9j35 · 2024-03-20

**Q2-1 Originality-Novelty:** 3
**Q2-2 Correctness-Technical Quality:** 3
**Q2-5 Clarity Of Writing:** 3

**Q1 Summary And Contributions:**

This article considers the setting of asynchronous federated bandits, where at each instance, an agent from a set of agents pulls an arm, and all agents collectively utilize their obtained data to tackle the Multi-Armed Bandit (MAB) problem. In this context, the objective of the MAB is pure exploration with fixed confidence, which means finding arms within a certain range from the optimal arm with high probability while using as little sample complexity as possible. Additionally, due to communication reasons, there are some extra constraints on communication cost.

**Q2-3 Extent To Which Claims Are Supported By Evidence:**

3: Good: the main claims are supported by convincing evidence (in the form of adequate experimental evaluation, proofs, (pseudo-)code, references, assumptions).

**Q2-4 Reproducibility:**

3: Good: key resources (e.g. proofs, code, data) are available and key details (e.g. proofs, experimental setup) are sufficiently well-described for competent researchers to confidently reproduce the main results.

**Q3 Main Strengths:**

This paper investigates a very novel problem and proposes two algorithms to address the Multi-Armed Bandit (MAB) and LeaderBoard (LB) settings, respectively. It characterizes the efficiency of the algorithms from two dimensions: sample complexity and communication cost. This work is comprehensive both in theory and experimentation, and the paper is well-written.

**Q4 Main Weakness:**

I am not quite clear about the learning objective in the context of asynchronous federated bandits. Looking at previous research, such as by He et al. [2022] and Li et al. [2023], the primary learning objective appears to still be regret. I think that in this respect, the authors need to provide some concrete examples to illustrate its practical significance.

**Q5 Detailed Comments To The Authors:**

I have a question regarding the problem setting: does the actual objective here consist of minimizing the sample complexity as well as minimizing the communication cost? Also, does the term "asynchronous" imply that in each round, after an agent pulls an arm and receives feedback, it can choose whether to upload the data or not? Does this mean that some data, if deemed not very significant, can be chosen not to be uploaded and used?

**Q9 Complying With Reviewing Instructions:**

Yes

---

> ### Author Rebuttal · Authors · 2024-04-05
>
> We thank the reviewer for the positive comments on our paper's contribution and novelty and valuable suggestions to clarify the arguments in our work.
>
> **Q1:** I am not quite clear about the learning objective in the context of asynchronous federated bandits. Looking at previous research, such as by He et al. [2022] and Li et al. [2023], the primary learning objective appears to still be regret. I think that in this respect, the authors need to provide some concrete examples to illustrate its practical significance.
>
> **A1:** Thank you for the question. The learning objective of the federated fixed confidence pure exploration problem studied in this paper is to identify an $\epsilon$-optimal best arm with high probability, with communication cost and sample complexity being as low as possible.
>
> Here is a practical example. Let's consider a sequential experimental design problem, e.g., for drug discovery or chemical synthesis, where our goal is to identify an arm that is $\epsilon$-near optimal (i.e., chemical with desired properties) with high probability. In this problem, we are not concerned about cumulative regret (i.e., the quality of the chemicals tried during the online learning process); instead, we only care about whether we can find the optimal arm **in the end**, and the corresponding sample complexity and communication cost due to their expensive nature (see the introduction in [Hillel et al., 2013; R'eda, 2022; Du et al., 2021] for details). Additionally, each laboratory lacks samples (i.e., funding for resource) to complete the task individually, so we need to involve multiple labs to collaborate on the learning task. These requirements motivate people to study federated pure exploration problems. Besides, previous synchronous federated pure exploration algorithms assume every agent (i.e., lab) should participate in the exploration (i.e., do the experiment) in each round and the server can force all the agents to upload their data in synchronization rounds. This is impractical due to some agents may get offline (e.g., they run out of resources), and all other agents should wait until they get online (e.g., collect enough resources), this will significantly reduce the learning speed (see the introduction in [He et al., Li et al., 2022, Li et al., 2023] for details). Our asynchronous federated pure exploration algorithms can alleviate these assumptions: 1) Each agent can decide whether to participate in each round. Full participation isn't obligatory, thus accommodating temporarily offline agents; and 2) communication between each agent and the server is asynchronous and completely independent of other agents. Based on our discussion, we believe our asynchronous federated pure exploration algorithms are more practical than the previous synchronous federated pure exploration algorithms.
>
> We will add this example in the future version of our paper.
>
> **Q2:** I have a question regarding the problem setting: does the actual objective here consist of minimizing the sample complexity as well as minimizing the communication cost? Also, does the term "asynchronous" imply that in each round, after an agent pulls an arm and receives feedback, it can choose whether to upload the data or not? Does this mean that some data, if deemed not very significant, can be chosen not to be uploaded and used?
>
> **A2:** Thank you for the insightful questions.
>
> 1. Yes. We answer this question in **A1**.
>
> 2. Yes. In our setting, the server cannot compel agents to participate in communication rounds. Active agents have the discretion to decide whether to communicate with the server or not. In our algorithms, we enable the active agent to communicate with the server only when a communication event is triggered; otherwise, the active agent refrains from uploading its data to the server. Besides, in the \textbf{design of communication event} of Section 4, we also mention that some agents may possess data that has not been uploaded to the server when the algorithm is terminated.
>
> We will enhance our paper's presentation based on this discussion.
>
> Once again, we thank you for these helpful questions. They really help us improve the clarity of the paper.
>
> **Reference:**
>
> [1] Li, C., $\And$ Wang, H. (2021). Asynchronous Upper Confidence Bound Algorithms for Federated Linear Bandits.
> [2] He, J., Wang, T., Min, Y., $\And$ Gu, Q. (2022). A Simple and Provably Efficient Algorithm for Asynchronous Federated Contextual Linear Bandits.
> [3] Xu, L., Honda, J., $\And$ Sugiyama, M. (2017). Fully adaptive algorithm for pure exploration in linear bandits. arXiv: Machine Learning.
> [4] Hillel, E., Karnin, Z.S., Koren, T., Lempel, R., $\And$ Somekh, O. (2013). Distributed Exploration in Multi-Armed Bandits.
> [5] R'eda, C., Vakili, S., $\And$ Kaufmann, E. (2022). Near-Optimal Collaborative Learning in Bandits.
> [6] Du, Y., Chen, W., Kuroki, Y., $\And$ Huang, L. (2021). Collaborative Pure Exploration in Kernel Bandit.

---

### Official Review · Reviewer_DWwV · 2024-03-21

**Q2-1 Originality-Novelty:** 3
**Q2-2 Correctness-Technical Quality:** 3
**Q2-5 Clarity Of Writing:** 3

**Q1 Summary And Contributions:**

The paper presents two federated asynchronous algorithms for multi-armed bandits and linear bandits respectively. The objective is to identify the best arm and they show upper bounds on the sample complexity for each algorithm. Also, they comment on how their complexity matches existing lower bounds and provide simulation experiments showing the performance of their algorithms.

**Q2-3 Extent To Which Claims Are Supported By Evidence:**

4: Excellent: all claims are supported by very convincing evidence (in the form of comprehensive experimental evaluation, rigorous mathematical proofs, detailed (pseudo-)code, precise references, well-motivated and realistic assumptions) and the authors deliver what they promise.

**Q2-4 Reproducibility:**

3: Good: key resources (e.g. proofs, code, data) are available and key details (e.g. proofs, experimental setup) are sufficiently well-described for competent researchers to confidently reproduce the main results.

**Q3 Main Strengths:**

The main strength of this paper is the level of technicality. This paper provides a complete analysis of their algorithms with matching upper and lower bounds. Also, it is generally well-written with clear section and subsection titles. It explains the arm selection strategies of their algorithms well.

**Q4 Main Weakness:**

The main weakness of this paper is perhaps the connection of this work to the general field or audience. It is somewhat hard to understand the level of contribution of their algorithms to the community as a non-expert in federated learning. Adding some discussion about e.g. the advantages of federated learning over standard BAI in bandits would be good.

**Q5 Detailed Comments To The Authors:**

- In experiments, it seems that the authors only compare their algorithms to single-agent and synchronous algorithms and it is no surprise that the proposed algorithm performs sub-optimally against them. Is there any asynchronous benchmark that the authors could compare to, say, e.g. asynchronous algorithms for regret minimization?

**Q9 Complying With Reviewing Instructions:**

Yes

---

> ### Author Rebuttal · Authors · 2024-04-05
>
> We thank the reviewer for appreciating our writing and presentation of the technical results. In the following we respond to the reviewer’s questions.
>
> **Q1:** The main weakness of this paper is perhaps the connection of this work to the general field or audience. It is somewhat hard to understand the level of contribution of their algorithms to the community as a non-expert in federated learning. Adding some discussion about e.g. the advantages of federated learning over standard BAI in bandits would be good.
>
> **A1:** Thank you for your valuable suggestion. Here, we outline the advantages of federated asynchronous pure exploration, which we will incorporate into revised version of our paper.
>
> 1. Federated pure exploration algorithms can accelerate the learning process. When employing single-agent pure exploration algorithms (such as UGapEc or LinGapE) independently on $M$ agents without communication, the sample complexity becomes $O(M)$ times larger than that of our algorithms. This suggests that **FAMABPE** can expedite the learning process by a factor of $O(M)$.
>
> 2. Federated pure exploration addresses challenges beyond the capabilities of single-agent pure exploration. Consider a scenario where individual agents lack sufficient samples (e.g., funding or resources) to accomplish fixed-confidence pure exploration tasks independently. In the federated pure exploration setting, by involving an adequate number of agents and utilizing federated pure exploration algorithms, we can effectively tackle such problems.
>
> 3. Our asynchronous algorithms offer higher practicality compared to their synchronous counterparts. Existing federated pure exploration algorithms are typically confined to synchronous settings [Hillel et al., 2013; R'eda, 2022; Du et al., 2021], wherein all agents are forced to upload their local data to the server upon request. Subsequently, agents download the latest data from the server after all uploads are completed. However, this requires full agent participation and global synchronization mandated by the server, making it impractical for many real-world application scenarios. In contrast, our asynchronous federated pure exploration algorithms can alleviate these constraints: (1) Each agent can decide whether to participate in each round. Full participation isn't obligatory, thus accommodating temporarily offline agents; and (2) communication between each agent and the server is asynchronous and completely independent of other agents. There's no need for global synchronization or mandatory coordination by the server.
>
> **Q2:** In experiments, it seems that the authors only compare their algorithms to single-agent and synchronous algorithms and it is no surprise that the proposed algorithm performs sub-optimally against them. Is there any asynchronous benchmark that the authors could compare to, say, e.g. asynchronous algorithms for regret minimization?
>
> **A2:** Thank you for your question. To the best of the author's knowledge, our proposed algorithms are the first to address pure exploration of asynchronous federated bandits. Additionally, existing works on regret minimization of asynchronous federated bandits [He et al., Li et al., 2022, Li et al., 2023] primarily concentrate on minimizing cumulative regret over $T$ iterations. In contrast, our algorithms provide $i_{ser,\tau}$ as the estimated best arm, while their algorithms lack a decision rule to output an estimated best arm. Therefore, we can not directly compare our results to theirs.
>
> **Reference:**
>
> [1] Hillel, E., Karnin, Z.S., Koren, T., Lempel, R., $\And$ Somekh, O. (2013). Distributed Exploration in Multi-Armed Bandits. ArXiv, abs/1311.0800.
> [2] R'eda, C., Vakili, S., $\And$ Kaufmann, E. (2022). Near-Optimal Collaborative Learning in Bandits. ArXiv, abs/2206.00121.
> [3] Du, Y., Chen, W., Kuroki, Y., $\And$ Huang, L. (2021). Collaborative Pure Exploration in Kernel Bandit. ArXiv, abs/2110.15771.
> [4] He, J., Wang, T., Min, Y., $\And$ Gu, Q. (2022). A Simple and Provably Efficient Algorithm for Asynchronous Federated Contextual Linear Bandits. ArXiv, abs/2207.03106.
> [5] Li, C., $\And$ Wang, H. (2021). Asynchronous Upper Confidence Bound Algorithms for Federated Linear Bandits. ArXiv, abs/2110.01463.
> [6] Li, C., Wang, H., Wang, M., $\And$ Wang, H. (2023). Learning Kernelized Contextual Bandits in a Distributed and Asynchronous Environment. International Conference on Learning Representations.

---

### Official Review · Reviewer_WD3P · 2024-03-22

**Q2-1 Originality-Novelty:** 3
**Q2-2 Correctness-Technical Quality:** 3
**Q2-5 Clarity Of Writing:** 3

**Q1 Summary And Contributions:**

This paper studies pure exploration in federated bandits with asynchronous communication, in which any agent can initiate a round of communication at any point in time. The paper considers both multi-armed bandits and linear bandits, and provides theoretical guarantees for both the sample complexity and communication costs.

**Q2-3 Extent To Which Claims Are Supported By Evidence:**

2: Fair: the main claims are somewhat supported by evidence (but the experimental evaluation may be weak, or does not match entirely with the claims, important baselines may be missing, proofs contain important ideas but lack rigor, algorithmic details are only discussed superficially, references are imprecise, assumptions are not sufficiently motivated or explicated, etc.).

**Q2-4 Reproducibility:**

3: Good: key resources (e.g. proofs, code, data) are available and key details (e.g. proofs, experimental setup) are sufficiently well-described for competent researchers to confidently reproduce the main results.

**Q3 Main Strengths:**

- The paper studies a novel problem setting in federated bandits, namely, pure exploration with asynchronous communication. Both multi-armed bandits and linear bandits are considered.
- It is particularly nice that the authors not only present the theoretical results, but also makes an effort to discuss the intuitions given by the theoretical results, such as Remark 1 and Remark 2.

**Q4 Main Weakness:**

- In Figure 1 (b), we cannot really see the advantage of the proposed method for linear bandits, because different from Figure 1 (a), the communication cost of the proposed method is also larger than the method with synchronous communication in most of the cases.
- If I understand correctly, for both multi-armed bandits and linear bandits, the algorithm results from incorporating pure exploration algorithms for standard bandits into federated bandits. So I think it would be good to clearly discuss what are the main resulting technical challenges in terms of both the design of the algorithms and the analysis.

**Q5 Detailed Comments To The Authors:**

I have discussed the strengths and weaknesses of the paper above.

**Q9 Complying With Reviewing Instructions:**

Yes

---

> ### Author Rebuttal · Authors · 2024-04-05
>
> We thank the reviewer for the positive comments on our paper's significance and novelty, and the constructive questions to help further clarify our idea and algorithms.
>
> **Q1:** In Figure 1 (b), we cannot really see the advantage of the proposed method for linear bandits.
>
> **A1:** Thank you for your insightful comment. We want to clarify that the asynchronous setting is intrinsically more difficult compared with its synchronous counterpart, which is also acknowledged in prior works studying regret minimization [He et. al., 2022; Li et al., 2021, 2023], i.e., asynchronous algorithms typically incur larger communication cost than synchronous ones under the same regret guarantee. Therefore, the inclusion of synchronous algorithms' mainly serves as a reference showing the performance under the easier synchronous setting. Specifically, the synchronous environment assumes that every agent participates in exploration in each round and that the server can initiate a global communication round. These assumptions ensure that the server receives global information in each synchronization round, enabling it to effectively exploit all data the agents have and achieve a lower sample complexity. However, in the asynchronous environment, such a global communication round is not feasible and there is no guarantee on when or whether an agent would become active. Hence, the server cannot achieve the global information unless the agents communicate with it in each round (which results in $2\tau$ communication cost).
>
> Besides, as highlighted in the introduction, the efficacy of existing federated synchronous pure exploration algorithms hinges on the strong assumptions of synchronous communication. In the case where these assumptions do not hold, e.g., due to existence of stragglers in real-world systems, synchronous communication becomes ineffective as it needs to wait for the slowest agent to respond. The main advantage of our **FALinPE** is its ability to achieve near-optimal performance in an asynchronous environment, a feat not achievable by any synchronous algorithms.
>
> We will add this discussion to the revised version of our paper.
>
> **Q2:** I think it would be good to clearly discuss what are the main resulting technical challenges in terms of both the design of the algorithms and the analysis.
>
> **A2:** Thanks for your comment. The main challenge addressed in our paper is to design federated pure exploration algorithms that can work in the asynchronous environment. We here highlight our contributions and the technical challenges.
>
> 1. A key challenge in conducting asynchronous pure exploration is the absence of dedicated synchronous communication rounds where the server can assign arms to be explored by each agent based on their latest observations. Moreover, there is no guarantee on when or whether an agent would become active again to execute the exploration and report its observations back. This severely hinders the applicability of all existing distributed/federated pure exploration algorithms, whose exploration strategies are based on solving optimal experimental design. To address this challenge, we adopt a fully adaptive exploration strategy, such that each agent separately and asynchronously decides which arm to pull, based on the statistics received from the server in its latest communication.
>
> 2. As discussed in the \textbf{design of the communication event} in Section 4, a technical challenge we addressed is that neither the agents nor the server have access to the global observation number $t$. Consequently, we cannot directly employ $t$ to establish exploration bonuses in the asynchronous environment. Hence, we design an upper bound for $t$ by leveraging the triggered event (i.e., event $2$ in the hybrid triggered strategy) to devise the exploration bonus. In Lemma $4$ and the proof of Lemma $2$, we demonstrate that $t \le (1+\gamma M)\sum_{k=1}^K T_{ser,t}(k)$. Subsequently, we substitute $(1+\gamma M)\sum_{k=1}^K T_{ser,t}(k)$ and $(1+\gamma M)\sum_{k=1}^K T_{m_t,t}(k)$ into the exploration bonuses of the server and $m_t$ to replace $t$.
>
> 3. The exploration bonuses in the linear bandit are not only related to $t$ but also related to covariance matrices. Therefore, the event-triggered communication protocol in the linear bandits is additionally required to keep $V_{m_t,t}$ and $V_{ser,t}$ in a desired proportion to the global covariance matrix $\lambda I + \sum_{s=1}^t x_{m_t,t}x_{m_t,t}^\top$. Based on this requirement, we propose a hybrid event-triggered strategy that can simultaneously control the size of $V_{loc,m,t}$ and $\sum_{k=1}^KT_{loc,m,t}(k)$. Our proof shows that the hybrid event-triggered communication protocol can also achieve a low communication cost compared with asynchronous regret minimization algorithms for linear bandits [He et al., 2022; Li et al., 2021, 2023].
>
> We will include the discussion in the revised version of our paper.
>
> **Reference:**
>
> See the reference in response to DWwV.

---

### Meta-Review · Area_Chair_Pcb3 · 2024-04-17

The paper examines the pure exploration problem in federated bandits under an asynchronous communication setting, covering both multi-armed and linear bandits. It extends traditional federated learning frameworks by introducing and analyzing the performance of the proposed algorithms in terms of sample complexity and communication cost.

Pros:
+ Introduces novel asynchronous algorithms for pure exploration in federated bandits.
+ Provides theoretical guarantees for both sample complexity and communication costs.
+ Discusses intuitions behind theoretical results, enhancing comprehension of the proposed methods.
+ Empirical results support the effectiveness of the proposed algorithms, although some aspects could be better highlighted.

Cons:
- The experimental results, especially for linear bandits, do not clearly demonstrate the advantages of the proposed methods over synchronous counterparts.
- The connection between standard pure exploration algorithms in federated settings and the new challenges introduced by asynchronous communication could be better elaborated.
- Some technical details and rigorous proofs to support the claims are not thoroughly presented or are missing crucial comparisons with relevant benchmarks.


The reviewers are mostly satisfied with the responses that authors have
provided. I trust that the authors can address the weaknesses when preparing the
camera ready version.